# A role for spindles in the onset of rapid eye movement sleep

Mojtaba Bandarabadi [1,2,3,5], Carolina Gutierrez Herrera [1,4,5], Thomas C. Gent [1,4,5], Claudio Bassetti[1,2], Kaspar Schindler [1,2] & Antoine R. Adamantidis [1,2,4✉]

Sleep spindle generation classically relies on an interplay between the thalamic reticular nucleus (TRN), thalamo-cortical (TC) relay cells and cortico-thalamic (CT) feedback during non-rapid eye movement (NREM) sleep. Spindles are hypothesized to stabilize sleep, gate sensory processing and consolidate memory. However, the contribution of non-sensory thalamic nuclei in spindle generation and the role of spindles in sleep-state regulation remain unclear. Using multisite thalamic and cortical LFP/unit recordings in freely behaving mice, we show that spike-field coupling within centromedial and anterodorsal (AD) thalamic nuclei is as strong as for TRN during detected spindles. We found that spindle rate significantly increases before the onset of rapid eye movement (REM) sleep, but not wakefulness. The latter observation is consistent with our finding that enhancing spontaneous activity of TRN cells or TRN-AD projections using optogenetics increase spindle rate and transitions to REM sleep. Together, our results extend the classical TRN-TC-CT spindle pathway to include non-sensory thalamic nuclei and implicate spindles in the onset of REM sleep.

[1] Department of Neurology, Zentrum für Experimentelle Neurologie, Inselspital University Hospital Bern, Bern, Switzerland. [2] Department of Neurology, Sleep–Wake-Epilepsy Center, Inselspital University Hospital Bern, Bern, Switzerland. [3] Department of Biomedical Sciences, University of Lausanne, Lausanne, Switzerland. [4] Department of Biomedical Research, University of Bern, Bern, Switzerland. [5] These authors contributed equally: Mojtaba Bandarabadi, Carolina Gutierrez Herrera, Thomas C. Gent. ✉email: antoine.adamantidis@dbmr.unibe.ch

During non-rapid eye movement (NREM) sleep, spindles emerge from thalamocortical interactions as transient and distinct brain oscillations (9–16 Hz) in humans[1–3], sheep[4], cats[5–7], rats[8–10], and mice[11–14]. In humans, spindles are defined as waxing and waning oscillations of variable peak amplitude (~100 μV) and duration (>400 ms) often coinciding with the UP-state of cortical slow waves and often occurring after a K-complex[15–17]. In humans, slow (~12 Hz) and fast (~14 Hz) spindles predominate in frontal[18,19] and centro-parietal neocortex[2,7], respectively. Consistent with scalp electroencephalography (EEG) recordings, fMRI studies confirmed the activation of distinct brain regions associated with slow and fast spindles[20], suggesting two different mechanisms of generation and offering the possibility of distinct functions. Spindle-rich EEG during NREM sleep—sometimes considered as distinct stage of NREM sleep[21,22]—has been proposed to reflect sleep stability in rodents and human[14,23,24], and often precedes REM sleep transition, yet, the implication of spindles in REM sleep onset remains unclear. Furthermore, a correlative role for spindles in memory consolidation, intelligence, and cognition has been proposed[25–29], and further confirmed by increased motor memory consolidation upon enhancement of spindle activity in humans[30], suggesting that the integrity of spindles is essential to higher brain functions including cognition. Yet, their organization in time and space is essential since optogenetic induction of spindle activity during wakefulness alters sensory attention in mice[31]. Accordingly, temporally extended and compressed spindles are associated with mental retardation and schizophrenia, respectively[32,33].

At the cellular level, spindles result from transient burst firing of thalamic reticular nucleus (TRN) neurons and thalamo-cortical (TC) relay cells, which generate typical spindling activity within cortico-TC pathways[10,13,14,16,34]. At the synaptic level, volleys of inhibitory inputs from TRN neurons in the frequency band of spindles provide a strong inhibition onto TC relay cells. The resulting inhibitory postsynaptic currents in TC cells evoke a large hyperpolarization that triggers a hyperpolarization-activated current, immediately followed by an activation of a T-type calcium current that results in a rebound burst of action potentials in TC cells upon termination of inhibitory postsynaptic currents[35]. Sensory TC cells excite layer IV pyramidal neurons in their corresponding cortical areas where they elicit excitatory postsynaptic potentials and spindle oscillations. Pacemaker activity of TRN cells[13], as well as functional inputs to TRN cells from cortical origins are sufficient to generate spindle-like activity[36], whereas cortical inputs to the TRN are responsible for spindle termination[37]. Similarly, recent dual recordings in humans suggest that convergent cortical DOWN states lead to thalamic DOWN states and thalamic cell hyperpolarization, hence triggering spindles that propagate to the cortex at the DOWN-to-UP state transition[34]. Accordingly, high temporal resolution fMRI in humans revealed a strong activation (increase in BOLD signal) of several thalamocortical structures including the lateral and posterior thalami, anterior cingulate (CING) cortex, insula, and other neocortical structures concomitant to single spindles[20]. In rodents, experimental recordings have classically focused on paired recordings between the TRN–TC–CT neurons in sensory thalamocortical pathways, mostly in anesthetized preparations[10,38–40].

Here, using multisite local field potential (LFP) and single-unit recordings in both primary sensory and non-sensory thalamic nuclei and cortical sites, we studied region-specific correlation of spindles with single-unit firing, slow waves, and vigilance state transitions in spontaneously sleeping (non-anesthetized) freely behaving mice. We further show that optogenetic activation of TRN[VGAT] neurons evoked spindles in connected somatosensory and anterior thalamic pathways, and increased transitions to rapid eye movement (REM) sleep.

## Results

We recorded simultaneous EEG, LFP, and single-unit activity from multiple thalamic and cortical sites across vigilance states in freely behaving mice using two configurations. In one set of experiments, we conducted LFP/unit recordings from the TRN, the ventrobasal complex of the thalamus (VB), the barrel cortex (BARR), and two EEGs and muscle tone (EMG) activity from freely moving mice ($n = 6$ animals; Supplementary Fig. 1; see "Methods" section). In the other one, we recorded LFP/unit recordings from the centromedial (CMT) and the anterodorsal (AD) thalamic nuclei, and the BARR, the CING and visual (VIS) cortices, as well as cortical EEGs and EMG from freely moving mice ($n = 12$), as described previously[41]. We detected spindles using a wavelet-based method[42,43], which showed superiority over bandpass-filtering approaches[18]. However, instead of the complex Morlet function used in previous studies[42–44], we incorporated the frequency B-spline function into the algorithm as an optimal wavelet function, and detected spindles from LFP/EEG recordings of freely behaving mice (Fig. 1; see "Methods" section).

On average, the algorithm detected $3.0 \pm 0.03$ min$^{-1}$ spindles from all the studied sites during NREM sleep in mice, consistent with previous studies in mice, rats, cats, sheep, and humans[1–5,8,11]. It also detected $0.25 \pm 0.03$ and $0.88 \pm 0.13$ min$^{-1}$ spindle-like events during wakefulness and REM sleep, respectively, which correspond to movement-related activity and phasic REM sleep (Supplementary Fig. 2)[45,46]. We then measured several parameters from regionally detected spindles in mice including the central frequency, duration, cycles, peak-to-peak amplitude, and symmetry[47] (Supplementary Fig. 3). In total, we analyzed $5457 \pm 73$ spindles per recording site for the CMT–AD–CING–BARR–VIS configuration, and $2588 \pm 11$ spindles for the other TRN–VB–BARR preparation. The central frequency of detected spindles was generally homogenous, from $11.3 \pm 0.02$ Hz for TRN to $11.7 \pm 0.02$ Hz for CMT, and equally distributed spatially with no distinguishable clusters of slow and fast spindles, consistent with previous studies in mice[12].

**Extending spindle pathways to non-sensory thalamocortical circuits.** To investigate cellular mechanisms that underlie regionally detected spindles, we estimated the spike–field coupling using the normalized cross-correlation between LFP and single-unit recordings during spindles from multiple thalamic and cortical sites (Fig. 2; see "Methods" section). We found that TRN single-unit activity is strongly phase-locked to LFP spindle cycles ($0.88 \pm 0.02$, $n = 12$ cells; Fig. 2a, c), as reported in the previous studies[5,10,13]. Interestingly, we further found that spike–field coupling within the CMT ($0.87 \pm 0.01$, $n = 15$ cells) and AD ($0.81 \pm 0.02$, $n = 10$ cells) thalamic nuclei is as strong as for TRN during spindles (TRN vs. CMT: $P > 0.99$; TRN vs. AD: $P = 0.18$; $F = 162.6$; d.f. $= 6$; one-way ANOVA with Tukey's post-hoc test; Fig. 2a, c). Time-lag analysis between single-unit activity and LFP spindle cycles revealed very short spike-field lags within TRN, CMT, and AD, with the shortest lag within the TRN (spike-field lag: TRN $= 4.3 \pm 0.8$ ms, CMT $= 9.1 \pm 0.6$ ms, AD $= 13.4 \pm 1.2$ ms; Fig. 2d). Indeed, TRN, CMT, and AD neurons are silent during the trough, and highly active during peak, of a spindle cycle.

Spike–field coupling during spindles also exists within CING ($0.69 \pm 0.01$, $n = 9$ cells), BARR ($0.63 \pm 0.02$, $n = 14$ cells), and VB ($0.62 \pm 0.01$, $n = 7$ cells), but significantly less than TRN (TRN vs. CING, BARR and VB: $P < 0.0001$; $F = 162.6$; d.f. $= 6$; one-way ANOVA with Tukey's post-hoc test; Fig. 2a–c). CING and VB neurons only showed increased firing during peaks (spike–field lag: CING $= 4.8 \pm 0.6$ ms, VB $= 5.3 \pm 0.8$ ms), although some neuronal activity remains during the troughs of the spindle cycles (Fig. 2a, b).

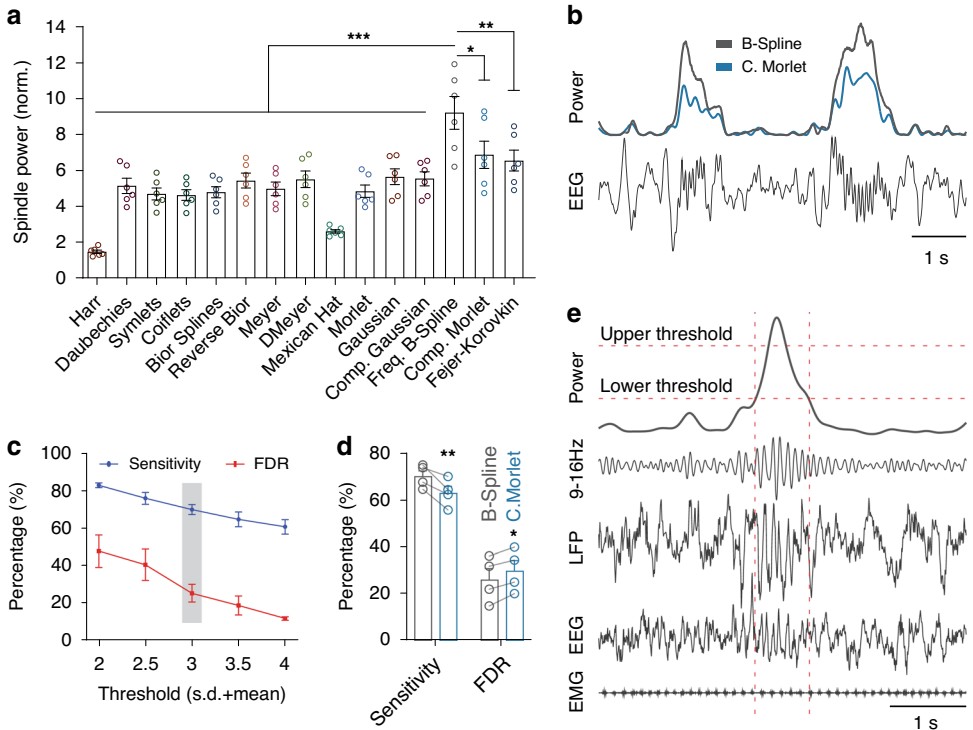

**Fig. 1 Improving the wavelet-based spindle detection method. a** Comparison between 15 different wavelet families to find the optimum function. Normalized spindle power indicates ratio between average wavelet energy of spindle segments and spindle-free segments. The complex frequency B-Spline wavelet function has significantly a higher normalized power as compared to other functions (complex frequency B-spline vs. other functions: $P < 0.0001$; $F = 14.22$; d.f. $= 14$; one-way ANOVA with Tukey's post-hoc test; *$P < 0.05$, **$P < 0.01$, ***$P < 0.001$; $n = 6$ subjects). **b** Representative spindles of human EEG signals and normalized wavelet energy within the spindle range (9–16 Hz) using the complex Morlet and frequency B-Spline functions. **c** Evaluation of the spindle detection algorithm on EEG recordings from naturally sleeping mice. Visual inspection of the automatically detected spindles by a human expert using different threshold levels revealed a sensitivity of 70 ± 2.7% and a false detection rate (FDR) of 25.2 ± 4.8% using the selected threshold (3SD + mean). Sensitivity and FDR are the number of correct and false detections, respectively, divided by sum of correctly detected and missed spindles. **d** Comparison between the complex frequency B-Spline and the complex Morlet functions for detection of spindles from EEG recordings of mice. Using the complex frequency B-Spline function as the core of the detection method provides significantly higher sensitivity and lower FDR (Sensitivity: $P = 0.002$; FDR: $P = 0.043$; two-way ANOVA with Sidak's post-hoc test; *$P < 0.05$, **$P < 0.01$; $n = 4$ animals). **e** Representative thalamic LFP and EEG/EMG signals of a detected spindle in mice. The dashed horizontal lines indicate upper and lower thresholds to detect the spindle and its start/end times, respectively. The dashed vertical lines indicate start and end of the spindle. Error bars indicate mean ± SEM. Source data are provided as a Source Data file.

BARR cells showed an inverse coupling, with highest neuronal spiking during the trough of cycles (spike–field lag: BARR = 37 ± 0.6 ms). Neuronal activity in the VIS cortex was weakly coupled to LFP fluctuations during spindles (0.2 ± 0.02; $n = 8$ cells), suggesting that detected events in LFPs are either generated in other cortical layers or resulted from volume conduction. These results first show the cellular substrates of local spindle oscillations in regions with high spike–field coupling and rule out false detection due to volume conduction or common referencing, and secondly, they extend the classical TRN–VB–BARR spindle pathway to other thalamocortical circuits.

**Spindles coincide with slow waves within the CMT, CING, and AD nuclei.** Previous studies reported a temporal coupling between slow waves (SW) and spindles in cortical LFP/EEG recordings of both human and animal models[15–17,48]. To test whether this coupling also exists in thalamic nuclei, we estimated SW–spindle correlation for all the recorded sites (Fig. 3). We first extracted SW activity (0.5–4 Hz) and spindle envelope, and then aligned them to the onset of spindles and averaged over all spindles of NREM sleep (Fig. 3a; see "Methods" section). We quantified the SW–spindle coupling using the normalized cross-correlation between the average traces explained above, and found a significant SW–spindle coupling within the CMT, CING,

and AD as compared to the TRN, VB, BARR, VIS, and EEGs (CMT, CING, and AD vs. others: $P < 0.05$; $F = 19.2$; d.f. $= 8$; one-way ANOVA with Tukey's post-hoc test; $n = 6$ animals for TRN/VB, $n = 12$ for other sites; Fig. 3b).

We further detected individual UP states, as described previously[41], and calculated the percentage of spindles that coincide with UP states of SWs (Fig. 3c; see "Methods" section). Similar to the SW–spindle coupling reported above, the CMT, AD, and CING showed significantly a higher ratio of spindles coincided with UP states as compared to the other sites (CMT, CING, and AD vs. others: $P < 0.01$; $F = 40.2$; d.f. $= 8$; one-way ANOVA with Tukey's post-hoc test; $n = 6$ animals for TRN/VB, $n = 12$ for other sites; Fig. 3c). These results suggest that SW–spindle coupling within non-sensory thalamic nuclei is stronger than for BARR and VIS, and that the CMT may contribute to SW–spindle coupling through modulation of SWs, which we reported recently[41].

To investigate the dependency of SW–spindle coupling on SW power, we calculated the correlation between SW–spindle coupling and delta power (0.5–4 Hz) during spindles for the thalamic and cortical sites (Fig. 3d; see "Methods"). We found a weak correlation for the thalamic nuclei, but a significantly high correlation for the cortical sites ($P = 0.079$ for thalamic nuclei; $P = 0.006$ for cortical sites; two-tailed Pearson's correlation test; $n = 6$ animals for TRN/

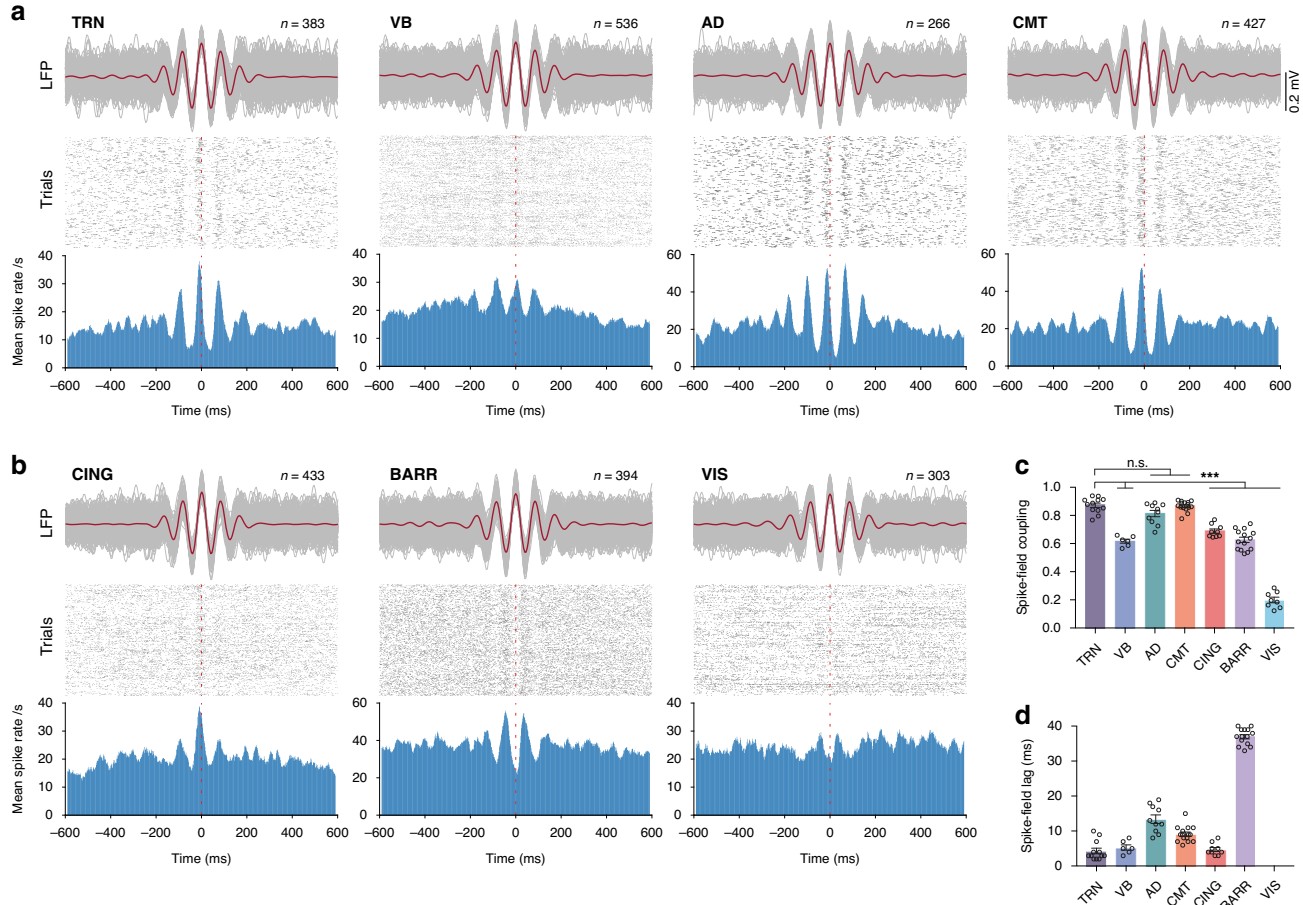

**Fig. 2 Cellular substrates of regional sleep spindles in thalamic nuclei and cortical sites. a** Representative average (red) and individual (gray) traces of thalamic spindles filtered within the spindle range (9–16 Hz) and aligned to peaks of central cycles of spindles. Each gray line represents one spindle, and $n$ indicate the number of spindles. Raster plots show single-unit activity of thalamic neurons during spindles, where each trial represents one spindle. The average neuronal firing rate of trials were obtained using a moving window of 10 ms having 80% overlap. Zero time indicates peaks of central cycles of spindles. **b** Same as **a**, but for cortical LFP/unit recordings. **c** Quantification of spike–field coupling during spindles using the normalized cross-correlation between filtered LFPs and average spike rate traces. Spike–field coupling within CMT and AD nuclei is as strong as in TRN (TRN vs. CMT: $P > 0.99$; TRN vs. AD: $P = 0.18$; $F = 162.6$; d.f. = 6; one-way ANOVA with Tukey's post-hoc test; ***$P < 0.001$; TRN: $n = 12$, VB: $n = 7$, AD: $n = 10$, CMT: $n = 15$, CING: $n = 9$, BARR: $n = 14$, VIS: $n = 8$ cells; 6 animals for TRN/VB and 12 for other sites). **d** Time lags between LFPs and single-unit activities of panel **c**. Error bars indicate mean ± SEM. Source data are provided as a Source Data file.

VB, $n = 12$ for other sites). When estimating the delta power during spindles for the first 30 min of recovery sleep, we found that delta power during spindles significantly increases for cortical, but not for thalamic, LFP recordings during recovery sleep as compared to baseline (recovery vs. baseline: $P < 0.001$ for CING and VIS; $P < 0.05$ for BARR; $F = 17.5$; d.f. = 8; two-way ANOVA with Sidak's post-hoc test; baseline: $n = 6$ animals for TRN/VB, $n = 12$ for other sites; recovery: $n = 4$ animals; Fig. 3e). These results suggest that SW power modulates both incidence and SW–spindle coupling in the cortical, but not in the thalamic, sites.

**Spindle rate significantly increases before REM sleep, but not wakefulness.** Spindles predominantly occur during NREM stage 2 in humans[2], yet their temporal distribution in the brain of spontaneously sleeping mice, which have only one defined NREM stage, remains unclear. Thus, we next examined the distribution of spindles in time and space across vigilance states in mice (Fig. 4a). We quantified spindle rate during NREM episodes followed by a transition to either wakefulness (N2W) or REM sleep (N2R), and different time scales ranging from 40 to 5 s before the termination of NREM sleep episodes and the onset of either wake or REM sleep (Fig. 4b; Supplementary Fig. 4).

We found that the spindle rate during N2R episodes is significantly higher than N2W episodes, with the highest rate occurring within a 25-s window prior to REM sleep onset, and up to $2.8 \pm 0.1$ times higher spindle rate as compared to N2W episodes (N2R and 25 s before REM vs. N2W: $P < 0.0001$ for all recording sites; $F = 891.2$; d.f. = 2; two-way ANOVA with Bonferroni's post-hoc test; $n = 6$ animals for TRN/VB, $n = 12$ for other sites; Fig. 4b; Supplementary Fig. 4). This increase before vigilance state transition was specific to REM sleep since N2W transitions showed no significant change in spindle activity (5–40 s before wake onset vs. N2W: $P > 0.99$ for all sites; $F = 531$; d.f. = 17; two-way ANOVA with Bonferroni's post-hoc test; $n = 6$ animals for TRN/VB, $n = 12$ for other sites; Supplementary Fig. 4a). Further analysis revealed an increase in the duration and number of cycles for the cortical LFP spindles in N2R as compared to N2W episodes (25 s before REM vs. N2W: duration: $P < 0.05$ for CING, BARR, and VIS; $F = 40.4$; d.f. = 1; number of cycles: $P < 0.05$ for CING, BARR, and VIS; $F = 39.2$; d.f. = 1; two-way ANOVA with Bonferroni's post-hoc test; $n = 6$ animals for TRN/VB, $n = 12$ for other sites; Supplementary Fig. 4c). In addition, we found no correlation between REM bout duration and spindle rate when calculated for the entire NREM episode

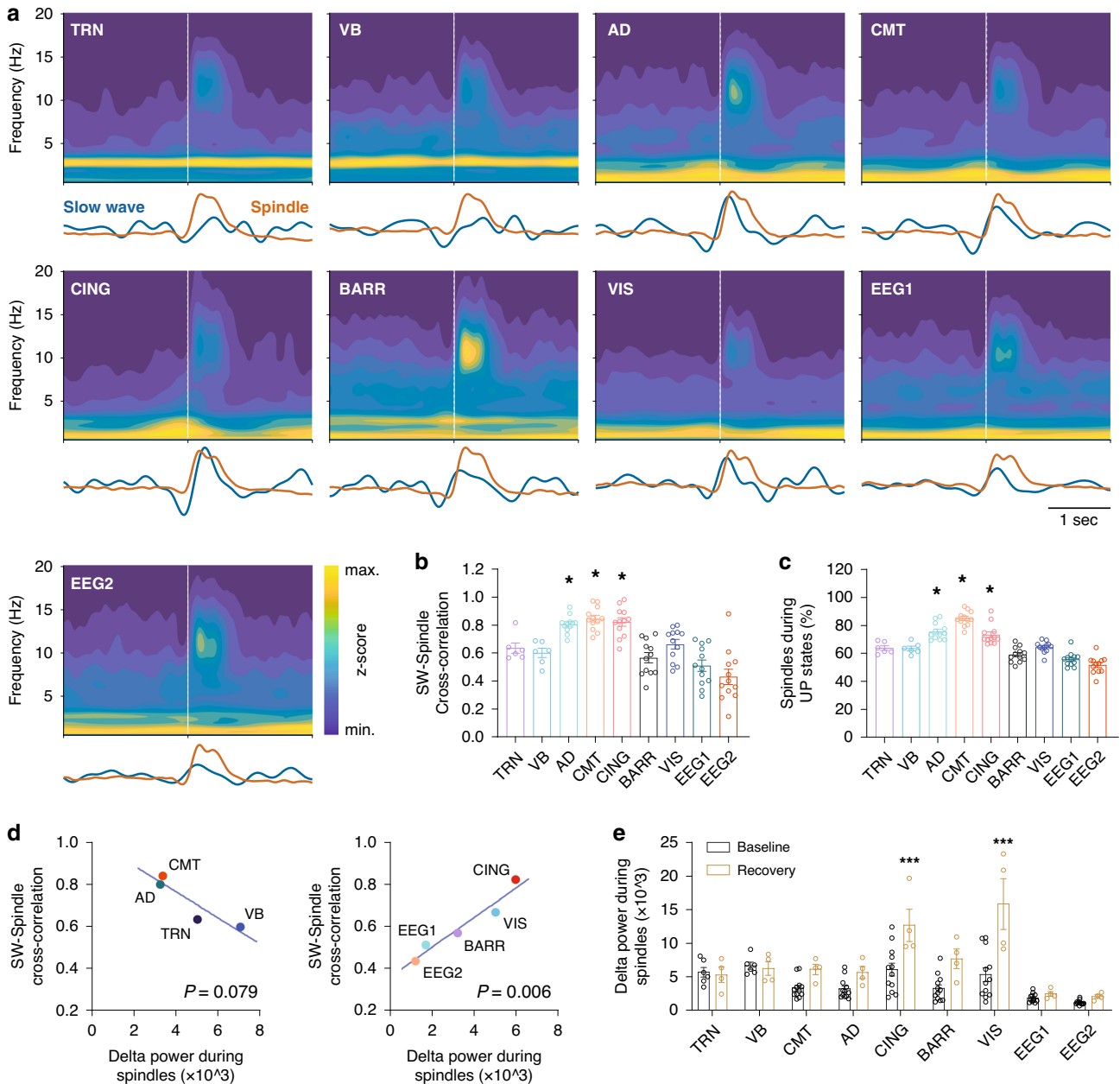

**Fig. 3 Spindles coincide with slow waves (SW) within the CMT, CING, and AD. a** Graphs show the average time–frequency representations of spindles. Vertical dotted white lines indicate the onset of spindles. Traces below the spectrograms show average of filtered LFPs for SW (blue, 0.5–4 Hz) and average of envelope of filtered LFPs in the spindle range (red, 9–16 Hz). Representative time–frequency graphs and traces were obtained from 334 ± 19 thalamic and 318 ± 39 cortical spindles. **b** Normalized cross-correlation between SW and spindle envelope indicates a significant SW–spindle coupling within the CMT, CING, and AD as compared to other sites (CMT, CING, and AD vs. others: $P < 0.05$; $F = 19.2$; d.f. = 8; one-way ANOVA with Tukey's post-hoc test, $*P < 0.05$; $n = 6$ animals for TRN/VB, $n = 12$ for other sites). **c** Percentage of spindles that coincide with UP states. The CMT, AD, and CING have significantly higher ratios of spindles occurred during UP states as compared to the other sites (CMT, CING, and AD vs. others: $P < 0.01$; $F = 40.2$; d.f. = 8; one-way ANOVA with Tukey's post-hoc test, $*P < 0.05$; $n = 6$ animals for TRN/VB, $n = 12$ for other sites). **d** Correlation between delta power during spindles and SW–spindle coupling for the thalamic and cortical sites. This correlation is weak for the thalamic nuclei, while significantly high for the cortical sites ($P = 0.079$ for thalamic nuclei; $P = 0.006$ for cortical sites; two-tailed Pearson's correlation test; $n = 6$ animals for TRN/VB, $n = 12$ for other sites). **e** Delta power during spindles significantly increases for cortical, but not for thalamic, LFP recordings during recovery sleep (yellow) as compared to baseline (black; recovery vs. baseline: $P < 0.001$ for CING and VIS; $P < 0.05$ for BARR; $F = 17.5$; d.f. = 8; two-way ANOVA with Sidak's post-hoc test, $***P < 0.001$; baseline: $n = 6$ animals for TRN/VB, $n = 12$ for other sites; recovery: $n = 4$ animals). Bars indicate mean ± SEM. Source data are provided as a Source Data file.

duration (N2R) or the 25 s before NREM-to-REM sleep transitions (Supplementary Fig. 5).

An increased power in the delta band activity is one of the major features of the sleep homeostatic process that compensate for sleep loss[49]. To study the effects of this process on spindles, we quantified the spindle dynamics during recovery sleep following a 4-h total sleep deprivation. We first performed the same analysis as for spontaneous sleep recording but we restricted our analysis to the first three REM episodes of recovery sleep. We found a similar pattern of increase in

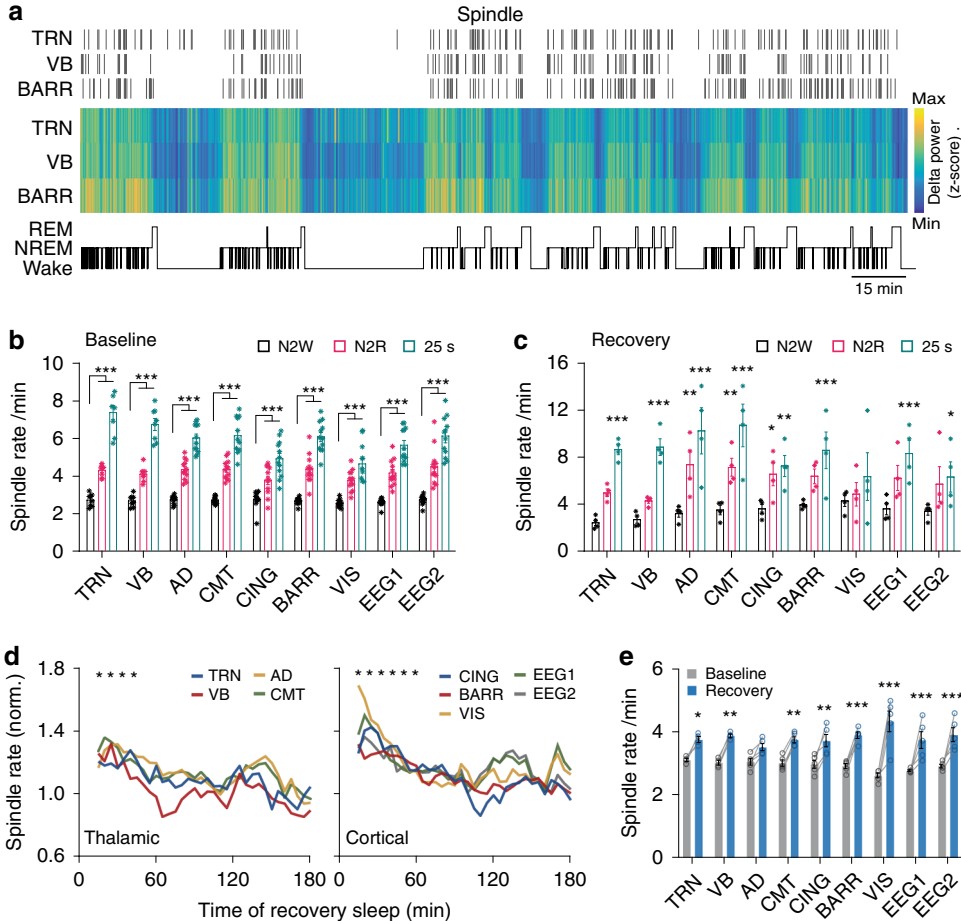

**Fig. 4 Spindle rate significantly increases before REM sleep, but not wakefulness. a** Representative spatiotemporal distribution of spindles across the vigilance states. Raster plot illustrates the detected spindles. Color-coded graph shows z-scored delta power across vigilance stats. Hypnogram is depicted below. **b** Spindle rate during NREM episodes before a transition to wake (N2W) or REM sleep (N2R), and within 25 s before REM sleep onset. Spindle rate of N2R episodes is significantly higher than of N2W, with a maximum rate within a window of 25 s before REM onset (N2R and 25 s before REM vs. N2W: $P < 0.0001$ for all sites; $F = 891.2$; d.f. = 2; two-way ANOVA with Bonferroni's post-hoc test; ***$P < 0.001$; $n = 6$ animals for TRN/VB and $n = 12$ for other sites). **c** Same as **b**, but for recovery sleep. Spindle rate significantly increases before transition to REM sleep (25 s before REM vs. N2W: $P < 0.001$; $F = 70.7$; d.f. = 2; two-way ANOVA with Bonferroni's post-hoc test; *$P < 0.05$, **$P < 0.01$, ***$P < 0.001$; $n = 4$ animals). **d** Dynamics of spindle rate during recovery sleep after 4 h of sleep deprivation, estimated using a moving window of 30 min having 25 min overlap. Spindle rate during recovery sleep significantly increased as compared to baseline, returning to baseline values after 40 min for the thalamic and 60 min for the cortical sites (recovery vs. baseline: thalamic nuclei: $P < 0.001$; $F = 5.9$; d.f. = 34; cortical sites: $P < 0.001$; $F = 10.4$; d.f. = 34; two-way ANOVA with Bonferroni's post-hoc test; *$P < 0.05$; $n = 4$ animals for TRN/VB and $n = 5$ for other sites). **e** Spindle rate during baseline and the first 30 min of recovery sleep for the individual sites showed that spindle rate significantly increased for all sites, except the AD. Cortical recordings showed the highest increase in spindle rate during first 30 min of recovery sleep (first 30 min of recovery vs. baseline: $P < 0.001$; $F = 190.7$; d.f. = 1; two-way ANOVA with Bonferroni's post-hoc test; *$P < 0.05$, **$P < 0.01$, ***$P < 0.001$; $n = 4$ animals for TRN/VB, $n = 5$ for other sites). Bars indicate mean ± SEM. Source data are provided as a Source Data file.

spindle rate before transition to REM sleep (25 s before REM vs. N2W: $P < 0.001$; $F = 70.7$; d.f. = 2; two-way ANOVA with Bonferroni's post-hoc test; $n = 4$ animals; Fig. 4c; Supplementary Fig. 4b). We then calculated spindle rate using a large moving window of 30 min with 25 min overlap to obtain a robust quantification. Interestingly, spindle rate during recovery sleep significantly increased as compared to baseline, and returned to baseline values after 40 min for the thalamic and after 60 min for the cortical sites (recovery vs. baseline: thalamic nuclei: $P < 0.001$; $F = 5.9$; d.f. = 34; cortical sites: $P < 0.001$; $F = 10.4$; d.f. = 34; two-way ANOVA with Bonferroni's post-hoc test; $n = 6$ animals for TRN/VB, $n = 12$ for other sites; Fig. 4d). Comparison of the spindle rate between baseline and first 30 min of recovery sleep showed that spindle rate significantly increased from all the recorded sites, except AD (first 30 min of recovery vs. baseline: $P < 0.001$; $F = 190.7$; d.f. = 1; two-way ANOVA with Bonferroni's post-hoc test; $n = 6$

animals for TRN/VB, $n = 12$ for other sites; Fig. 4e). On average cortical recordings showed a higher increase in spindle rate during first 30 min of recovery sleep (Fig. 4d, e).

**Driving TRN[VGAT] cells or their projections to AD increases spindle rate and the probability of transition to REM sleep.** The TRN has been identified as the main pacemaker of spindles in sensory thalamocortical circuits[13,14,50]. Yet, our spike–field coupling results indicate cellular substrates of spindles within non-sensory thalamus. Additionally, the spatiotemporal distribution of spindles suggests a possible function of spindles in the regulation of NREM-to-REM sleep transitions. To further investigate these two findings, we induced spindles by optogenetic activation of TRN[VGAT] cells or their terminals within the AD nuclei of the thalamus (AD) in freely moving mice[51–53]. In contrast to previous studies[13,14], we targeted the expression of the stabilized

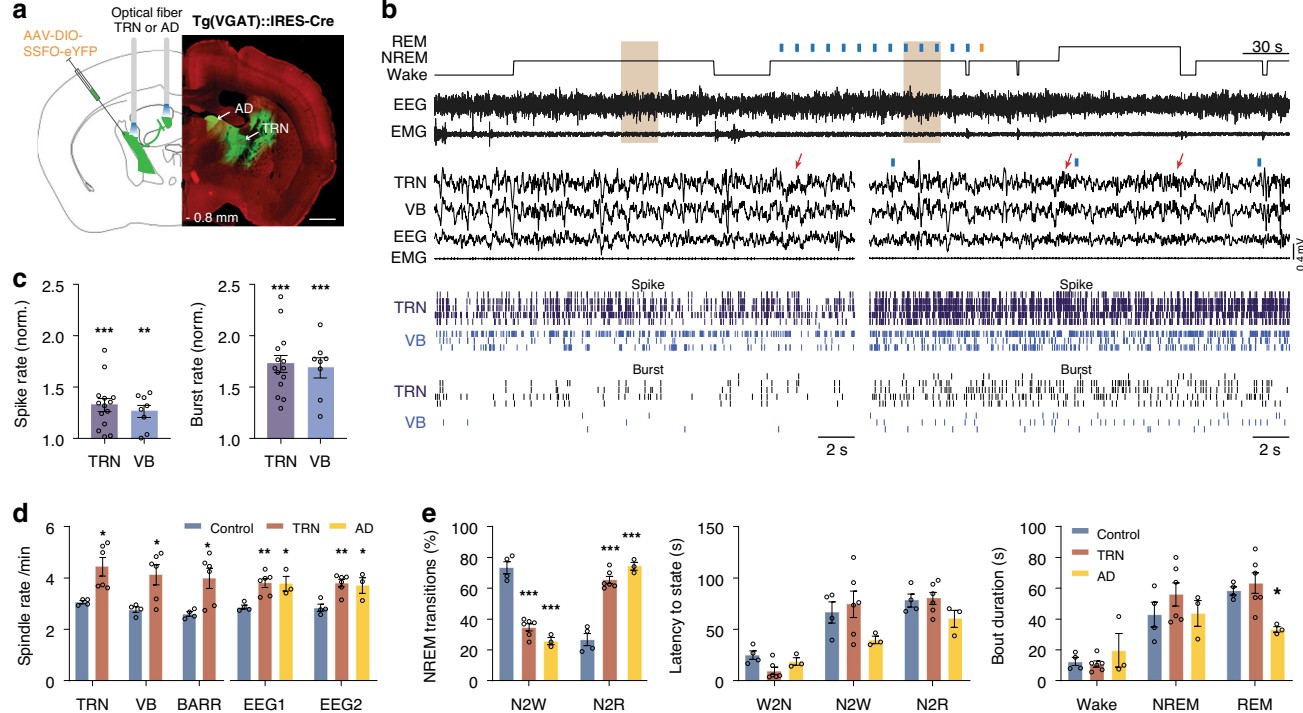

**Fig. 5 Optogenetic activation of TRN^VGAT cells or TRN-AD terminals increases spindle rate and probability of transitions to REM sleep. a** Schematic representation of stereotactic injection of a EF1α-DIO-SSFO-EYFP adeno-associated virus in the anterior TRN of VGAT::IRES-Cre driver mice, and bilateral optic fiber placement in TRN or AD. **b** Representative EEG/EMG and thalamic LFP/unit/burst activities during NREM sleep episodes outside and within stimulation of TRN^VGAT cells. The brown-shaded areas in the top traces are expanded below. Red arrows indicate the detected spindles. **c** Quantification of single-unit and burst firing of TRN ($n = 14$ cells) and VB ($n = 8$ cells) neurons during TRN activation, normalized to the baseline condition. Both single-unit and burst firing of TRN and VB neurons significantly increased as compared to baseline (TRN stimulation vs. control: TRN: unit: $P = 0.0002$; $t = 5.1$; d.f. = 13; burst: $P < 0.0001$; $t = 8.8$; d.f. = 13; $n = 14$ cells, 6 animals; VB: unit: $P = 0.003$; $t = 4.4$; d.f. = 7; burst: $P = 0.0002$; $t = 7.0$; d.f. = 7; $n = 8$ cells, 6 animals; two-sided paired $t$-test; **$P < 0.01$, ***$P < 0.001$). **d** Spindle rate significantly increases upon optogenetic activation of TRN^VGAT neurons or their terminals within AD (TRN stimulation vs. control: TRN: $P = 0.032$, VB: $P = 0.035$, BARR: $P = 0.041$, EEG1: $P = 0.004$, EEG2: $P = 0.009$; AD stimulation vs. control: EEG1: $P = 0.013$, EEG2: $P = 0.016$; $F = 4.48$; d.f. = 4; two-way ANOVA with Tukey's post-hoc test; *$P < 0.05$, **$P < 0.01$; $n = $ control:4, TRN:6, AD:3 animals). **e** Averaged NREM sleep transitions (left), latency to next states (middle) and bout duration upon SSFO activation of TRN (red), AD (yellow), or control (blue; TRN or AD stimulation vs. control: NREM transitions: $P < 0.0001$ for N2R and N2W; $F = 0.82$; d.f. = 2; latency to vigilance state: $P > 0.05$ for all states; $F = 1.7$; d.f. = 4; two-way ANOVA with Tukey's post-hoc test; *$P < 0.05$, **$P < 0.01$, ***$P < 0.001$; $n = $ control: 4, TRN:6, AD:3 animals). All values are reported as mean ± SEM. Source data are provided as a Source Data file.

step-function opsin (SSFO) to TRN neurons, and their terminals located in AD, by stereotactic injection of a EF1α-DIO-SSFO-EYFP or EF1α-DIO-EYFP (control) adeno-associated virus (AAV) in the anterior TRN of VGAT::IRES-Cre driver mice (Fig. 5a; see "Methods" section). We used SSFO to induce non-synchronous natural activity of targeted cells[52], instead of ChR2 activation that induce synchronized action potentials upon trains of light pulses. We chronically implanted bilateral optical fibers dorsal to the TRN or AD in distinct set of animals, and recorded LFP/unit from TRN, VB, BARR, and EEG/EMG signals for the TRN stimulation, and only EEG/EMG signals for the AD stimulation, in freely moving mice (Fig. 5a, b; Supplementary Fig. 1; see "Methods" section).

We then optogenetically activated SSFO-expressing TRN^VGAT cells, or their terminals within AD, during randomly selected NREM sleep episodes by delivering 50-ms blue light pulses every 10 s and then closing the channel with single 100-ms yellow light pulse upon vigilance state transition. We found that activation of SSFO-expressing TRN cells significantly increased both single-unit and burst firings of TRN and VB neurons as compared to non-stimulated conditions (TRN stimulation vs. non-stimulated: TRN: unit: $P = 0.0002$; $t = 5.1$; d.f. = 13; burst: $P < 0.0001$; $t = 8.8$; d.f. = 13; $n = 14$ cells, 6 animals; VB: unit: $P = 0.003$; $t = 4.4$; d.f. = 7; burst: $P = 0.0002$; $t = 7.0$; d.f. = 7; $n = 8$ cells, 6 animals;

two-sided paired $t$-test; Fig. 5b, c). Furthermore, optogenetic activation of SSFO-expressing TRN^VGAT neurons, or their terminals within AD, significantly increased spindle rate as compared to control conditions (TRN stimulation vs. control: TRN: $P = 0.032$, VB: $P = 0.035$, BARR: $P = 0.041$, EEG1: $P = 0.004$, EEG2: $P = 0.009$; AD stimulation vs. control: EEG1: $P = 0.013$, EEG2: $P = 0.016$; $F = 4.48$; d.f. = 4; two-way ANOVA with Tukey's post-hoc test; $n = $ control:4, TRN: 6, AD: 3 animals; Fig. 5d).

Consistent with our correlative data, state-specific optogenetic activation of SSFO-expressing TRN^VGAT neurons, or TRN^VGAT-AD projections, during NREM sleep significantly increased the probability of NREM-to-REM transitions without affecting REM sleep latency (TRN or AD stimulation vs. control: NREM transitions: $P < 0.0001$ for N2R and N2W; $F = 0.82$; d.f. = 2; latency to vigilance state: $P > 0.05$ for all states; $F = 1.7$; d.f. = 4; two-way ANOVA with Tukey's post-hoc test; $n = $ control: 4, TRN: 6, AD: 3 animals; Fig. 5e). Note that REM sleep episode duration is slightly reduced upon TRN-AD projection activation as compared to TRN activation or controls, suggesting an implication of TRN-AD circuit in transition rather than maintenance of REM sleep (AD stimulation vs. control: $P = 0.022$; $F = 2.9$; d.f. = 4; two-way ANOVA with Tukey's post-hoc test; $n = $ control: 4, AD: 3 animals; Fig. 5e).

## Discussion

Using an optimized spindle detection method, we showed the presence of NREM sleep spindles as discrete events in thalamic/cortical LFPs, spiking patterns and EEG recordings from spontaneously sleeping mice. These spindles share similarities with those reported in other species[54], suggesting evolutionary-conserved mechanisms and functions. We found that neuronal spiking patterns in the thalamus and neocortex of mice are strongly phase-locked to the spindle LFP activity, and that spindles occur in thalamic nuclei outside of the classical TRN–VB–BARR pathway, particularly the CMT and AD nuclei of the thalamus. We further showed that optogenetic activation of TRN$^{VGAT}$ neurons, or TRN$^{VGAT}$-AD projections, is sufficient to induce spindles and facilitate NREM to REM sleep transitions. These later results are consistent with our finding that spindle-rich EEG during NREM sleep predicts the onset of REM sleep, as previously suggested[21,22]. Of note, gender-specific effects of the "spindle–REM sleep" relationship due to high spindle rate described in female rodents as compared to males[55] warrant further investigation.

We showed for the first time a very strong spike–field coupling within non-sensory thalamic nuclei (CMT and AD) during spindles in addition to the previously reported TRN and sensory VB nucleus, which form a classical spindle pathway within thalamus[10,56,57]. As the LFP measures the synaptic inputs of a large number of neurons[58], existence of spike–field coupling during spindles strongly suggests that local neurons contribute to spindle occurrence. Thus, our findings provide causal evidence for the extension of the classical somatosensory TRN–VB–BARR pathway supporting spindles propagation within the thalamocortical networks. We observed a reversed spike–field coupling within the layer V of BARR cortex. A possible explanation is that the layer VI, which provides strong glutamatergic cortico-thalamic feedback[59,60], is the cortical pacemaker of spindles, and that cortical layers V and VI are extracellular current sinks and sources, respectively, as shown using current source density analysis[61,62]. Similar reversed cortical spike–field coupling during spindles was also observed in rats[63].

Locally detected spindles showed a strong coupling to the regional SWs within the CMT, CING, and AD, which was weaker for the TRN, VB, BARR, VIS, and cortical EEGs. In a recent study[41], we identified a thalamocortical pathway for the initiation and propagation of global SWs, consisting of CMT, CING, AD, and VIS. We showed that the CMT neuronal firing entrains cortical spiking activity in the CING and then the SW propagates to the VIS through the thalamic relay of AD nucleus, and proposed that CMT neuron population strongly modulates frontal SWs[41]. This may explain the strong SW–spindle coupling within the CMT, CING, and AD nuclei that also exhibited potent spike–field coupling during spindles, suggesting that they are functional circuit nodes underlying spindle and SW pathways in thalamocortical circuits during NREM sleep. This coupling was weaker for the VIS as compared to the other nodes of the SW propagation pathway. A possible explanation is that the detected spindles within VIS also did not show spike–field coupling, indicating they might be due to volume conduction.

Anatomically, sensory thalamic nuclei (VB) have almost no direct projections to non-sensory midline thalamus nuclei including the CMT. Yet, our study clearly shows the existence of spindles in these nuclei (CMT, AD). Therefore, strong spike–field coupling within these nuclei suggests a separate circuit for spindle generation that may generate spindles independent of the classical TRN–VB–Barrel cortex circuit. Spindles occurring in these separate networks may have different functional significance that directly relate to the region-specific organization of sleep structure[64,65], and sleep functions including the coordination of somatosensory (VB) information and attention or executive functions (CMT, AD)[38,66–69].

We also found spindle-like events in the thalamus and neocortex during wakefulness and REM sleep, with lower rates than NREM sleep. These events share common oscillatory patterns with NREM spindles in LFP/EEG signals, but they essentially originate from fast theta activity during wakefulness (e.g., locomotion) or phasic REM sleep[45,46]. The individual and averaged time–frequency representations of detected spindle-like events explicitly show that they occur during sustained periods of theta activity (6–10 Hz), resulting from a sudden short increase in the frequency of theta oscillations. Whether these spindle-like events during wakefulness and REM sleep have similar origins and pathways as NREM spindles requires further investigation.

Previous studies induced spindles by stimulation of the TRN in mice[13,14,70], and cortex in humans[30]. Here, we used optogenetic to confirm the role of the TRN as one of the drivers of sleep spindles and to study its effect on NREM–REM sleep architecture. However, instead of stimulating the TRN at a spindle relevant frequency as in the previous studies[13,14,70], we employed SSFO to increase TRN cell excitability and connected networks rather than imposing artificial spindle-range optogenetic stimulation. Using SSFO, we increased the spontaneous generation of action potentials of a subgroup of TRN$^{VGAT}$ cells. Interestingly, while single-unit firing rates increased by only ~30%, the bursting activities increased by ~70% and was associated with an increase in spindle rate in connected thalamocortical areas including VB, BARR, and cortical EEGs. Of note, single blue light pulses did not systemically induce spindles, suggesting that other TRN inputs, such as cortico-thalamic inputs, or the overall state of the thalamocortical network, might be required for the full generation of a spindle as they occur during spontaneous NREM sleep.

Our results show that the onset of REM sleep directly depends on the distribution of sleep spindles in time and space, in addition to stabilizing NREM sleep[14,24] or supporting sleep-dependent memory consolidation[27,28,70,71]. Indeed, optogenetic activation of either TRN cell bodies, or their terminals within AD, both facilitate the onset, but not the latency, of REM sleep. A possible explanation is that the onset of REM sleep requires a highly stabilized NREM sleep, which is often accompanied by high activity in the frequency range of spindles (or sigma band)[24]. Accordingly, our data showed that the highest probability of spindles occurs within a time window of ~25 s prior to REM sleep onset. Interestingly, this time window coincides with the infra-slow variations (0.02 Hz) of the power of the sigma band[24], considering that half cycle of infra-slow waves is ~25 s. This window may correspond to the descending phase of infra-slow waves, previously suggested to be associated with the termination of a NREM sleep episode[24]. This is further confirmed by our observation that spindle rate is significantly increased during recovery sleep in mice, consistent with the dynamics of slow, but not fast, spindles in humans[72]. Considering similar central frequencies of spindles in mice and slow spindles in humans, their analogous dynamics during recovery sleep may have the same origin and a facilitatory role in REM sleep onset.

In conclusion, we assessed spindles in both LFP recordings and single-unit activity from multisite thalamic and cortical sites. Our results showed that spindles in non-sensory thalamocortical circuits originate from local synchronous neuronal activities, in addition to the classical sensory thalamic nuclei. We further showed that spindle dynamics change during NREM to REM sleep transitions reflecting a deepening of sleep, suggesting a role of spindles in the regulation of sleep–wake states.

## Methods

**Animals**. We used C57BL6 male mice from Charles Rivers Laboratories, Germany and Tg(VGAT)::IRES-Cre mice. Animals were housed in individual custom-designed polycarbonate cages at constant temperature (22 ± 1 °C), humidity

(30–40%), and circadian cycle (12-h light–dark cycle, lights on at 08:00). Food and water were available ad libitum. Animals were treated according to protocols and guidelines approved by the Veterinary office of the Canton of Bern, Switzerland (License number BE 113/13). Only adult (>6 weeks old) male mice were used in the experiments. Animals were housed in IVC cages in groups of 2–5 before instrumentation and after virus injections. After implantation, all mice were housed individually. Animals were habituated to the recording cable and optical fibers in their open-top home cages (300 × 170 mm) and kept tethered for the duration of the experiments. Animals were allowed to move freely in the cage during in vivo electrophysiology experiments. Before commencing experimental recording, baseline sleep was recorded and compared to previously published results[52], to confirm resumption of a normal sleep–wake cycle such as percentage and episode duration, after chronical instrumentation (Supplementary Fig. 6). Experiments were performed during the "lights-on" period (12:00–17:00). Viral injections were performed at 6 weeks of age and instrumentation at 10 weeks of age. All recordings were performed from 11 to 14 weeks of age. Note that mouse data with CMT–AD–CING–BARR–VIS configuration were recorded for another study[41].

**Stereotaxic injection of AAV**. Six-week-old C57BL6 or Tg(VGAT)::IRES-Cre mice were anaesthetized in isoflurane (4.0% for induction, 1.0–1.5% for maintenance) in oxygen and mounted in a stereotaxic frame (Model 940, David Kopf Instruments). Saline 10 ml/kg and meloxicam 5 mg/kg were given subcutaneously before incision. The skin on the head was shaved and aseptically prepared, and lidocaine 2 mg/kg infused subcutaneously at the incision site. A single longitudinal midline incision was made from the level of the lateral canthus of the eyes to the lambda skull suture. Injections were performed using a 28-gauge needle (Plastics One) connected by mineral oil filled tubing to a 10 μl Hamilton syringe in an infusion pump (Model 1200, Harvard Apparatus). Injections were performed in TRN and AD at 0.05 μl/min and the needle left in situ for 10 min afterwards to allow diffusion. For TRN and AD activation, 300 nl of AAVdj carrying EF1α-DIO-SSFO-EYFP or control Ef1α-DIO-EYFP vector were injected per site. Plasmids were kindly provided by Dr. K. Deisseroth, virus vectors were packaged at the Vollum Vector Core, University of Washington. Animals were left for 3–4 weeks before instrumentation or sacrifice for histology.

**Instrumentation**. Animals were anaesthetized in isoflurane in oxygen and mounted in a stereotaxic frame. Saline 10 ml/kg and meloxicam 5 mg/kg were given subcutaneously. The skin on the head was shaved and aseptically prepared, and lidocaine 2 mg/kg infused subcutaneously at the incision site. A single longitudinal midline incision was made from the level of the lateral canthus of the eyes to the lambda skull suture. Two stainless steel screws were placed in the skull to measure EEG (EEG1: AP −1.5 mm, ML +2.0 mm, EEG2: AP −2.8 mm, ML +2.5 mm) and two bare-ended wires sutured to the trapezius muscle of the neck to record EMG. Tetrodes were made from four strands of 10 μm twisted tungsten wire, connected to an electrode interface board by gold pins and were inserted into the TRN (AP −0.6 mm, ML +1.5 mm, DV −3.5 mm), VB (AP −1.7 mm, ML −1.82 mm, DV −3.6 mm), AD (AP −0.9 mm, ML +0.8 mm, DV −3.2 mm), CMT (AP −1.7 mm, ML +1.0 mm, DV −3.8 mm, 15°), CING (AP +1.8 mm, ML +0.2 mm, DV −1.6 mm), BARR (AP −2.0 mm, ML +2.2 mm, DV −1.1 mm), VIS (AP −3.3 mm, ML +2.5 mm, DV −0.9 mm), and secured to the skull with dental acrylic (C&B Metabond). Optic fibers of 200 μm diameter were placed in the TRN (AP −0.6 mm, ML +1.5 mm, DV −3.5 mm) or AD (AP −0.9 mm, ML +0.8 mm, DV −3.2 mm) and secured with the same dental acrylic. Finally, the implant was stabilized using a methyl methacrylate cement and the animal allowed a minimum of 5 days to recover in the home cage on top of a heating mat before starting recordings.

**In vivo electrophysiological recording**. For all recordings, mice were connected to a tethered digitizing headstage (RHD2132, Intan Technologies) and data sampled at 20 kHz recorded in free open source software (RHD2000 evaluation software, Intan Technologies). Optical fibers were connected to a patch chord using a zirconia sleeve (Doric Lenses). The connection was covered in black varnish to prevent ocular stimulation from the laser. Habituation to the cables was performed up to 8 h per day until the animals had nested and resumed a normal sleep–wake cycle. All baseline recordings, recovery sleep, and optogenetic experiments were performed between ZGT4 and 9. For sleep deprivation, gentle handling was performed when animals were stationary to prevent sleep between 8:00 (ZGT 0) and 12:00 (ZGT 4), and EEG/EMG and LFP/unit recordings were obtained during recovery sleep between ZGT 4 and 9.

Optogenetic stimulation was performed with 473 nm blue light from a laser (LRS-0473-PFM-00100-05, Laserglow Technologies) via a patch chord, 10 s after the onset of NREM sleep, judged by an experienced experimenter in real time as described previously[53]. Optical inhibition was performed with 593 nm yellow light from a laser (LRS-0532-GFM-00100-03, Laserglow Technologies) also via a patch chord. SSFO experiments activation was initiated using 50 ms of 3 mW blue light delivered every 10 s throughout the duration of NREM sleep episode. Termination of the activation was done by delivering 100 ms pulse of 10 mW yellow light as reported previously[52]. Laser output was controlled using TTL from a pulse generator (Master-9, AMPI or PulsePal 2, Sanworks). TTL signals were co-acquired

with all recordings. To obtain LFP, EEG, and EMG signals, the raw recordings were downsampled to 1000 Hz after applying a low pass filter (Chebyshev Type I, order 8, low pass edge frequency of 400 Hz, passband ripple of 0.05 dB) to prevent aliasing.

**Immunohistochemistry**. Animals were deeply anaesthetized with 15 mg pentobarbital (i.p.) and the heart transfused with 20 ml ice cold heparinized PBS followed by 30 ml 4% formalin. Brains were removed and post-fixed overnight in 4% formalin. They were then cryoprotected in 40% sucrose for 24–48 hr. Sections of 30 μm were cut in a cryostat. Free-floating sections were washed in PBS plus 0.1% Triton X-100 (PBS-T) three times for 10 min each and then blocked by incubation with 4% bovine serum albumin in PBS-T for 1 h. Free floating sections were incubated with primary antibodies for GFP (Life Technologies: A10262; 1:4000) for 24–48 h at 4 °C. They were then washed in PBS-T, three times for 10 min each and then incubated with secondary antibody (Abcam: AB96947, 1:500) for 1 h at room temperature. Confirmation of electrode placement was performed in brain sections stained with bisBenzimide as a counter stain to the DIO that coated the electrodes. Briefly, free-floating brain slices were exposed to bisBenzimide (1 μg/ml) in PBS for 15 min at room temperature. Three washes of 15 min each in PBS were then performed and slices then mounted on glass slides and allowed to dry. A cover slip was placed on the slices with a mounting medium and then imaged on a confocal fluorescent microscope.

**Determination of vigilance state**. We scored vigilance states manually, blind to the experimental conditions, in 1 s epochs using the concurrent evaluation of EEG and EMG signals. Wake episodes cover periods of either theta band EEG activity and EMG bursts of movement-related activity, or periods that mice were immobile including feeding and grooming behaviors. We scored NREM sleep as periods with a relatively high amplitude low-frequency EEG and reduced muscle tone relative to wakefulness associated with behavioral quiescence. We scored REM sleep as sustained periods of theta band EEG activity and behavioral quiescence associated with muscle atonia, save for brief phasic muscle twitches. Supplementary Fig. 6 shows statistics of vigilance states including percentage, episode duration, and density.

**Data analysis**. Data analyses were carried out using custom scripts written in MATLAB® (R2018b, MathWorks, Natick, MA, USA). Furthermore, built-in functions from Wavelet and Signal Processing toolboxes of MATLAB were investigated.

**Detection of spindles using an optimal wavelet function**. We detected spindles using the wavelet-based method proposed in ref. [42], which showed superiority over bandpass filtering approaches[18]. However, instead of the complex Morlet function used in the above study, we first screened for an optimal wavelet function for spindle detection. We applied continuous wavelet transform on human EEG signals from the DREAMS sleep spindle dataset[73] using 15 different wavelet families (Fig. 1a). The DREAMS spindle dataset consists of central scalp EEG channels recorded from eight human subjects, and two human experts have independently marked the spindles during NREM stage 2 sleep of these recordings. Two subjects from the DREAMS dataset have been annotated by only one expert and were excluded from our analysis. We first estimated wavelet energy in the frequency band of spindles (9–16 Hz), and then ranked wavelet functions using ratio between average wavelet energy of spindle segments and spindle-free segments. A higher normalized wavelet energy during spindles indicates a higher correlation between a wavelet function and spindle patterns. We also tuned free parameters of wavelet functions (order, bandwidth, and central frequency) using a grid search to further optimize wavelet functions.

We then developed our spindle detection algorithm using wavelet energy of the complex frequency B-spline function, which provided the highest normalized power, by considering several criteria. The wavelet energy time series was smoothed using the 200 ms Hann window, and a threshold equal to 3 SD (SD: standard deviation) above the mean was applied to detect potential spindle events. Afterward, we set a lower threshold of 1 SD above the mean to find start and end of detected events. Events shorter than 400 ms or longer than 2 s were discarded. Using bandpass-filtered LFP signals in the spindle range (9–16 Hz), we automatically counted the number of cycles of each event and excluded those with <5 cycles or more than 30 cycles. To ensure that increase in wavelet energy is spindle specific, and it is not due to artefacts, we estimated power in the spindle range as well as 6–8.5 and 16.5–20 Hz frequency bands, and discarded those events that power within the spindle band was lower than of two other bands. We also estimated the central frequency of spindle using the Fourier transform. We measured the symmetry of spindles using the position of peak of wavelet energy time-series with regard to the start and end of spindles. The symmetric measure lies between 0 and 1; 0.5 corresponds to complete symmetry, and values lower and higher than 0.5 show a leftward and rightward shift of peak, respectively.

To verify the efficiency of our spindle detection method, we measured the sensitivity and false detection rate (FDR) of the method on both human and mice EEG recordings. We found that when applied to the human DREAMS database[73], our algorithm had a sensitivity of 89.9 ± 3.4%, indicating a high detection

sensitivity. Furthermore, the algorithm detected $0.56 \pm 0.12$ spindles per min for human subjects that had not been visually scored, yet had sufficient criteria to be considered as spindles on further VIS inspection[18].

**Single unit analysis**. Multiunit activity was first extracted from bandpass-filtered recordings (600–4000 Hz, fourth-order elliptic filter, 0.1 dB passband ripple, $-40$ dB stopband attenuation). Filtering was performed in both the forward and reverse directions (filtfilt, MATLAB R2018b signal processing toolbox). The detection threshold was set as 7.5 times the median of the absolute value of the filtered signal. The detected multiunit activity was then sorted using the WaveClus toolbox[74]. Briefly, the four-level Haar wavelet transform was applied to the detected multiunit activity, and 10 most discriminative wavelet coefficients were selected using the Kolmogorov–Smirnov test. Selected wavelet coefficients were subsequently sorted using super-paramagnetic clustering to obtain single units. We visually inspected sorted spikes and excluded clusters with a completely symmetric shape, as noise clusters, or with a mean firing rate $< 0.2$ Hz from further analysis. Mean firing rate during NREM baseline and NREM SSFO experiments was calculated as total number of action potentials during each condition divided by total time spent in that state and reported as number per second (Hz). Burst firing of single units was detected as a minimum of three consecutive action potentials with inter-spike intervals (ISIs) $< 6$ ms, and preceded by a quiescent hyperpolarized state of at least 50 ms[75].

**Spike–field coupling**. We estimated coupling between unit activity and LFP signal, called spike–field coupling, by averaging unit activity during all regionally detected spindles, aligned to central peaks of regional spindles. To find central peak of a spindle, we first filtered LFP signal for the spindle range (9–16 Hz) using a 333rd order window-based finite impulse response (FIR) filter (fir1, MATLAB R2018b signal processing toolbox) in both the forward and reverse directions. We then considered central peak as closest peak of the filtered signal to the center of spindle. We calculated mean firing rate by averaging firing rates using a 10 ms moving window with 8 ms overlap. To quantify spike–field coupling, we calculated normalized cross-correlation between average of filtered LFP signals and mean firing rate during spindles.

**Correlation between slow waves and spindles**. We filtered LFP/EEG signals for SW (0.5–4 Hz) and spindle (9–16 Hz) frequency bands using the 6000th and 333rd order window-based FIR filters, respectively, in both the forward and reverse directions. We extracted envelope of spindles using the Hilbert transform. We then aligned both SW activity and spindle envelope to the start of spindles, detected by algorithm, and averaged across entire NREM spindles. To quantify correlation between SWs and spindles, we estimated normalized cross-correlation between average signals of SWs and spindle envelope. To find ratio of spindles that coincide with UP states, we detected UP states as reported previously[41]. Briefly, LFP/EEG signals were first bandpass filtered (0.5–4 Hz) using the 6000th order window-based FIR filter in both the forward and reverse directions. Individual UP states were detected from zero-crossing of filtered signals. The start and end of UP states were defined as zero-crossing from negative to positive and vice versa, respectively. To secure our analysis, we excluded individual UP states that were shorter than 200 ms or had absolute amplitude less than the absolute mean. We then estimated temporal overlap of a spindle with UP states, and considered a spindle coincide with an UP state if there exist at least 50% temporal overlap. We obtained coincident ratio as number of spindles co-occurred with UP states divided by total number of spindles for each recording site/animal.

**Quantification of spindle rate during vigilance state transition**. We estimated spindle rate before state switching from NREM to REM and wake, separately. We first marked all NREM–REM and NREM–Wake transition points, which were scored with 1 s resolution, and then calculated spindle rate using different time scales, ranging from 5 to 40 s with a 5-s incremental window, before vigilance state transition. We averaged over all transitions for each animal to obtain the spindle rate.

**Spectral analysis**. Power spectral density (PSD) was estimated using the Welch's method (pwelch, MATLAB R2018b Signal Processing Toolbox), using 8 s windows having 75% overlap. Delta power in a recording segment was calculated using a modified periodogram with Hanning window (bandpower, MATLAB). Delta power during spindles was estimated using the periodogram with a 4-s Hanning window centered on spindles. Time–frequency representations were obtained using continuous wavelet transform with the complex Morlet function.

**Statistical methods**. MATLAB® (R2018b, MathWorks, Natick, MA, USA) and Prism 8 (GraphPad) were used for statistical analysis. No power calculations were performed to determine sample sizes, but similarly sized cohorts were used in other relevant investigations[41,53]. Data were compared via one-way or two-way ANOVA followed by multiple comparisons tests and $t$ tests for parametric data, as indicated in the text. Values in the text are reported as mean ± standard

error mean (SEM) unless reported otherwise. Figures were prepared in Adobe Illustrator CC (Adobe).

**Reporting summary**. Further information on research design is available in the Nature Research Reporting Summary linked to this article.

## Data availability
The data that support this study is provided as Source Data. The DREAMS sleep spindle database is publically available online at https://zenodo.org/record/2650142#.X0k6Icgzbg4. Source data are provided with this paper.

## Code availability
The script for automatic detection and characterization of spindles, developed for this study, is available as the supplementary file.

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

## Acknowledgements

We thank the Tidis laboratory members for their technical help and comments on a previous version of the manuscript. We thank the laboratory of H.-R. Widmer and the M.I.C. UNIBE Facility for the use of the microscopes. Optogenetic plasmids were kindly provided by K. Deisseroth (Stanford University) and E. Boyden (MIT). M.B. was supported by the Inselspital University Hospital Bern, and Swiss National Science Foundation (190605). A.R.A. was supported by the Human Frontier Science Program (RGY0076/2012), Inselspital University Hospital, the University of Bern (IRC "Sleep & Health"), Swiss National Science Foundation (156156), and the European Research Council (ERC-2016-COG-725850).

## Author contributions

M.B., C.G.H., T.C.G., and A.R.A. designed the study. M.B., C.G.H., and T.C.G. collected and analyzed the data. A.R.A. supervised the project. M.B., C.G.H., T.C.G., C.B., K.S., and A.R.A. discussed the results and wrote the manuscript.

## Competing interests

The authors declare no competing interests.
