## [Peer Review File · Nature Communications]

Reviewers' comments:

Reviewer #1 (Remarks to the Author):

The manuscript by Bandarabadi, et al. uses savvy experimental neuroscience techniques and thoughtful analyses to address measures of local single unit-LFP coupling during spindles in several cortical and thalamic areas, examine cross-correlations between spindles and slow waves, optogenetically induce more spindles in TRN to increase REM duration, and determine spindle dynamics prior to REM or wake or even recovery from sleep deprivation.

While I appreciate the technically challenging nature of the work that was well executed, a main issue is that the sum of the findings is somewhat poorly tied together, and it almost seems as though it can be divided into 2 manuscripts, one that focuses more on coupling and one that focuses more on the relationship to REM sleep. The coupling findings, particularly the spike-field coupling findings, are novel, although I would argue that a topographically organized connectivity between the TRN and various thalamic subnuclei was described in Halassa, et al, Cell 2014. Additionally, the functional (or other) implications of the coupling observed in the different cortical and thalamic areas is not well described. The quantification of increased spindle density and other spindle parameters leading up to REM sleep is probably the most thorough I've encountered in the literature, although the idea of a spindle-rich transition to REM period (even as a distinct stage of rodent NREM sleep) has long been championed by the Gina Poe lab (e.g. Gross, et al, J. Neurosci. Methods, 2009; Watts, et al, J.Neurosci, 2012; Swift, et al, Curr.Biol. 2018 – see also Glin, et al, Physiology and Behavior, 1991). Use of the step function opsin to increase spindle density and thereby prolong REM duration causally implicates spindles in sleep stability and aligns with the observed increase in REM entries when spindle-like stimulation was optogenetically induced in mice (Kim, et al, PNAS, 2012) (although this group also saw increased stability of non-REM sleep, and additional compare/contrast of the authors' findings with this paper may be warranted).

Other minor issues include:

1. There is a bit of confusion on the validation of their spindle detecting algorithm. For the validation on the human dataset, the authors write that, "Furthermore, the algorithm detected 0.56 ± 0.12 spindles per minute for human subjects that had not been visually scored, yet had sufficient criteria to be considered as spindles on further visual inspection." This seems to suggest that these spindles were not ones expertly annotated by the DREAMS expert scorers, but were picked up by some other method – but then such a method would have its own sensitivity and FDI rates. I think listing the sensitivity and FDI rates against the gold standard expert scoring is sufficient without having to compare against other automated spindle detection algorithms (such as the Warby, et al, one listed). Also, when applied to the rodent dataset, what was considered the ground truth for rodent spindles? Were they also hand-scored by experts?
2. In figure 3a, can the authors define what the vertical dotted white line represents? It appears to be the onset of the spindle envelope, but this could be clarified.
3. Also related to figure 3a, why do the spectrograms show a prominent narrow band of slow wave activity around 3 Hz for TRN and VB, whereas it is much lower (0-1 Hz) in most of the other areas examined, including other thalamic areas AD and CMT? Is there something different about the slow waves in these areas?
4. The figure 3 legend lists a, b, c, and d panels, but only a-c are in the actual figure.
5. In the figure legend for Figure 4b, can the authors clarify that the brown-shaded areas in the top tracing are expanded in middle tracings below, with perhaps better labeling of the pre-stim vs intra-stim tracings. Also I assume the red arrows are highlighting spindles, but this could be stated explicitly.
6. In the recovery sleep experiments, the authors note that spindle density is increased. In the context of recovery sleep, do some of the earlier findings still apply? i.e. is spindle density equally elevated before REM during recovery sleep? Is the increased spindle density during recovery sleep predictive of longer REM bouts?
7. In the Methods under Stereotaxic Injection of AAV, there is no mention of the step function

opsin and its control (only ChETA). Also include virus titer and total volume injected.

8. In the legend for Figure 2, can the authors reiterate the number of animals that contributed to the unit pooling?

9. The authors used only male mice in these experiments. Given that females have increased spindle density compared to males, a discussion of the generalizability and potential effect of sex may be warranted.

10. The discussion of and potential explanation for reversed spike-field coupling within the BARR cortex is appreciated. A study in naturally sleeping rats found neocortical units fired at the trough of a spindle (trough-locked) which corresponded to a layer 4 sink and layer 5 source (Sirota et al, PNAS 2003), and similarities/differences to this finding may warrant discussion.

Reviewer #2 (Remarks to the Author):

The article by Bandarabadi, Gutierrez Herrera et al. describes properties of spindle oscillatory patterns in different thalamic nuclei and the neocortex in mice. With an innovative optogenetic stimulation protocol in TRN, they increase spindling during NREM sleep and find that this prolongs later REM sleep periods.

I enjoyed reading this paper. It argues well why multisite recordings in freely behaving mice, and naturalistic spindle stimulation protocols add value to the extant spindle related research (e.g. p. 19 l. 423ff). It was, however, hard for me to follow why this particular set of results was presented in one manuscript and how the different aspects contribute to a main takeaway.

The paper reports two sets of results, 1) spindle detection and properties in different thalamic nuclei and the neocortex. 2) The relation of spindling to sleep architecture (and effects of spindle stimulation on sleep architecture). It did not become entirely clear to me, why 1) is necessary to report, and how it supports the findings relating to 2). In my opinion, how spindles relate and maybe control sleep architecture is the more novel and exciting finding reported here.

I therefore believe that this manuscript could be made a stronger paper if the first part was e.g. moved to the supplement (if the authors feel the need to report evidence that validates their spindle detection method), or considerably shortened, and the authors focused instead on reporting results relating to objective 2). This would additionally give them more space for better theoretical embedding of that topic.

Comments:

A lot of space is given to description of the spindle detection method. It is not clear, however, whether the differences in this algorithm (wavelet based, and specifically the B-spline function) crucially contributed to the new findings, or whether the same pattern of results would have been observed with other methods? I personally think that robustness towards different analysis methods speaks to the robustness of an observed effect. It might not be necessary to stress the specifics of the used method in a lot of detail, if it is not at the center of the findings. I would find it helpful if the authors commented in the description of their method whether it merely detects spindles more precisely, or whether the detection method was actually crucial to their findings.

What were the controls for the optogenetic stimulation results? How exactly were the baseline values obtained? Can the authors rule out non-specific effects of light, heat, or protein expression?

Does the optogenetic stimulation protocol allow testing whether stimulation and thus increased spindling can promote/facilitate REM sleep onset as suggested by the authors given the results reported on spindle grouping relative to vigilance state transitions? This would help answer whether spindles have a causal role here or whether the timing of REM sleep is controlled by the

infraslow oscillations that in turn control frequency of spindle occurrence.

Spindle results in BARR: It could be made even more prominent that the inverse findings in BARR probably relate to the fact that an unusual layer was recorded, such that readers do not put too much focus/weight on the directionality of these findings (e.g. p. 8 l.185).

p.17 l.364 confirmed that TRN is a driver of spindles is a little misleading in context. The authors only show this for the classical TRN-VB-BARR pathway, not for the CMT and AD pathways the report as novel findings in part 1).

What is a possible significance of increased spindle rate during recovery sleep? How does this finding relate to the other results reported in this paper? It was hard for me to follow how this adds to the authors' message, it would be nice if more context were given.

What is a possible significance of the additional spindle pathways (CMT, AD) compared to the classical TRN VB? Discussion of this/directions for further research would strengthen the case of including the results.

Have the authors considered mu motor rhythm during wakefulness and alpha bursts during REM sleep as possible contributors to "falsely" detected spindle events in wake and REM where sleep spindles in the traditional sense are not expected?

Additional notes:

Note of caution: I would suggest using frequency of spindle occurrence rather than spindle frequency. Particularly in the human literature (which is also cited in the introduction) spindle frequency usually relates to the oscillatory frequency of a spindle (as in fast and slow spindles). When talking about both in the same manuscript, explicit specification may avoid confusion.

Sometimes, small inconsistencies in language made it difficult for me to follow the authors line of argumentation (e.g. p. 8 180-182, "but significantly less than of TRN", p. 10 l.222-225, p. 17 l.365-367 "... which increases in frequency").

Reviewer #3 (Remarks to the Author):

The authors present an impressive and novel paper showing that 1) areas of the thalamus that have not previously been linked with spindles show spindle coupling, and 2) spindles increase briefly before REM sleep, and stimulating spindles increases the length of ensuing REM. The paper is very well-written, statistically thorough, and the literature is well-covered. I do believe, however, that the paper could be improved by considering the points below.

Major considerations:

1) My largest issue is not one due to faulty methods or incorrect interpretations. It's simply that this paper feels like it has two separate ideas that do not connect well. The first section talks about evidence that thalamic nuclei that have not been previously linked with spindles show spindle coupling; the second section uses optogenetics to stimulate spindles and shows an interesting connection with REM length as a result. But the authors do not record from (or do not report on) the same non-specific areas in the second section, making me wonder why these are not separate papers. To be clear, I believe the second section is interesting enough on its own to warrant publication here (provided the other reviewers do not raise substantial concerns), whereas the first is more fitting for a more specialized journal, considering there is no novel contribution of what function these areas support. If the authors prefer to keep these as the same paper, I'm

wondering if they could skirt through the first section more quickly and expand upon the second section.

2) The optogenetic stimulations occurred every 10 seconds. One way to expand the second section is to ask more questions related to these stimulations, such as - were the stimulations any less effective at increasing spindles when they occurred shortly after preceding spindles (when the system has been proposed to be in refraction)? That is, do they differ as a function of the time since the preceding spindle? This could be important for understanding the characteristics of this refractory period and, in turn, spindle generation.

3) It appears stronger coupling between slow waves and spindles occurs in AD, CMT, and CING because of stronger slow wave power in these regions, as seen in figure 3A. The authors also briefly touch on this idea in the discussion without testing it explicitly. Could the authors verify (via quantification) whether this is indeed what drives the coupling difference?

4) Regarding the increase in spindles after sleep loss, it is well-known that slow waves increase after sleep loss. Is the spindle increase explained by this change in slow waves, or is it meaningfully independent?

5) In figure 5C, it is not clear what the baseline control values are for the first 40 or 60 minutes, and how this recovery night comparison to the baseline as a result. Please either plot this as a different score or plot the baseline night as well.

Minor considerations:

1) One impressive and understated aspect of this paper is the development of a new, superior spindle algorithm. If they have not already, the authors should consider sharing the code to this algorithm as part of the paper.

2) The author state, "Our findings, thus, provide causal evidence for the extension of the classical somatosensory TRN-VB-BARR pathway for propagation of spindles within thalamocortical circuits." Could the authors please explain how this claim is justified?

3) The author state, "The natural extension to this is that mice do not have centro-parietal spindles, i.e. predominantly TRN originating spindles." Could the authors please explain what they mean by this, using references to back up their claim?

4) Please fix this sentence: "We found that spindles waveforms were highly symmetrical in all the recorded sites."

5) Please fix this sentence: "The complex frequency B-Spline wavelet function has significantly a higher normalized power compared to other functions (frequency B-spline versus other functions: $P < 0.0001$; $F = 14.22$; d.f. = 14; one-way ANOVA; $n = 6$ subjects)."

Revision summary

Our study has obviously two sections: a first one, mainly computational, aiming at improving reliable detection of spindles in the brain of sleeping mouse, and a second one, more functional, that identifies a clear link between the occurrence of spindles during NREM sleep and the onset of REM sleep. As noted by all 3 reviewers, both parts provide new insights into cellular substrates and possible functions of spindles. Yet, we agreed with all three reviewers that both sections did seem “disconnected” and “not well balanced” in the previous version of the manuscript.

Thus, in the revised manuscript, we downsized the first section (including the coupling) while improving the second one with new data and new analysis that causally involved the TRN-AD circuit in NREM-to-REM sleep transition and further consolidate a role for spindles in the onset of REM sleep. Accordingly, we changed the title of the study to: “A role for spindles in the onset of rapid eye movement sleep”. Furthermore, we performed new analysis on our datasets, and indeed, we found that the increase of spindle density before REM during sleep rebound is (even) significantly higher than during baseline (Fig. 4c).

Finally, we thank all the reviewers for their insightful comments and suggestions that help to improve the revised manuscript and strengthened the conclusions of the study.

Reviewer #1

The manuscript by Bandarabadi, et al. uses savvy experimental neuroscience techniques and thoughtful analyses to address measures of local single unit-LFP coupling during spindles in several cortical and thalamic areas, examine cross-correlations between spindles and slow waves, optogenetically induce more spindles in TRN to increase REM duration, and determine spindle dynamics prior to REM or wake or even recovery from sleep deprivation.

Answer: We would like to thank the reviewer for the positive feedback on both the experimental and analysis parts of the study.

1. While I appreciate the technically challenging nature of the work that was well executed, a main issue is that the sum of the findings is somewhat poorly tied together, and it almost seems as though it can be divided into 2 manuscripts, one that focuses more on coupling and one that focuses more on the relationship to REM sleep.

Answer: We thank the reviewer for this relevant comment. As described above, we have improved the link between the two sections of our study and improve the functional significance of spindles in (REM) sleep architecture.

2. The coupling findings, particularly the spike-field coupling findings, are novel, although I would argue that a topographically organized connectivity between the TRN and various thalamic subnuclei was described in Halassa, et al, Cell 2014. Additionally, the functional (or other) implications of the coupling observed in the different cortical and thalamic areas is not well described.

Answer: We agreed that the previous study by Halassa et al (2014) reported connectivity of TRN with several subthalamic nuclei (including the visual-projecting and anterior-projecting TRN neurons) and provide important data to the field. The novelty of our work lies the extensive characterization of spindles from multiple thalamic and cortical sites (note that some people are still convinced that spindle do not exist in mice...) and their implications in sleep-wake state architecture, in particular NREM-to-REM sleep transitions, using multisite recording of neural activities in (freely-moving) spontaneously sleeping mice.

The present study was not designed to investigate the functional implications of spindles, which remains a very interesting question. However, we have included a discussion paragraph on a possible functional implication of the coupling observed in the different cortical and thalamic areas.

3. The quantification of increased spindle density and other spindle parameters leading up to REM sleep is probably the most thorough I've encountered in the literature, although the idea of a spindle-rich transition to REM period (even as a distinct stage of rodent NREM sleep) has long been championed by the Gina Poe lab (e.g. Gross, et al, J. Neurosci. Methods, 2009; Watts, et al, J. Neurosci, 2012; Swift, et al, Curr.Biol. 2018 – see also Glin, et al, Physiology and Behavior, 1991).

Answer: We recognized that the idea that spindle rich sleep often precede REM sleep is not novel, and gave credit to these qualitative studies in the revised manuscript. However, our work is the first to 1) provide a versatile spindle detection method and report a precise quantification of spindle occurrence during sleep; and 2) establish a causal link between the occurrence of spindles and the onset of REM sleep.

In addition to previous optogenetic activation of TRN neurons, we now provide new results/analysis describing that TRN^{GABA}-AD circuit activation is sufficient to increase NREM-to-REM sleep transition, suggesting that AD is one of possibly several targets supporting such transitions in the mammalian brain. Note that others subthalamic targets may exist, as discussed in the revised discussion section.

4. Use of the step function opsin to increase spindle density and thereby prolong REM duration causally implicates spindles in sleep stability and aligns with the observed increase in REM entries when spindle-like stimulation was optogenetically induced in mice (Kim, et al, PNAS, 2012) (although this group also saw increased stability of non-REM sleep, and additional compare/contrast of the authors' findings with this paper may be warranted).

Our new data further implicate both the TRN and its projections to AD in NREM-to-REM transitions. In addition to technical differences with the Kim et al study (2012), these data now suggest that there might be different thalamic circuit involved in such NREM-to-REM sleep transitions. We now described and discussed this point in light of the reviewer's comment (Kim et al 2012), as well as Luthi's studies on sleep fragility.

Other minor issues include:

1. There is a bit of confusion on the validation of their spindle detecting algorithm. For the validation on the human dataset, the authors write that, "Furthermore, the algorithm detected 0.56 ± 0.12 spindles per minute for human subjects that had not been visually scored, yet had sufficient criteria to be considered as spindles on further visual inspection." This seems to suggest that these spindles were not ones expertly annotated by the DREAMS expert scorers, but were picked up by some other method – but then such a method would have its own sensitivity and FDI rates. I think listing the sensitivity and FDI rates against the gold standard expert scoring is sufficient without having to compare against other automated spindle detection algorithms (such as the Warby, et al, one listed). Also, when applied to the rodent dataset, what was considered the ground truth for rodent spindles? Were they also hand-scored by experts?

Answer: This is a good point. To clarify, we believe that there is no real ground truth when scoring spindles in human or rodents, and this is precisely why we developed spindle detection algorithm by optimizing spindle detection in human, where spindles are easily identified by visual observations, before confronting our algorithm to rodent EEG/LFP recordings.

For the human data, all spindles from the human database were annotated by two independent human scorers (Devuyst, Dutoit et al. 2011). Yet, our algorithm identified few additional ones (not detected previously) that were further confirmed visually by a human expert from our laboratory.

For the rodent dataset, we used the spindle algorithm as the initial detection methods before confirmation by visual inspection. This is now clarified in the revised manuscript.

2. In figure 3a, can the authors define what the vertical dotted white line represents? It appears to be the onset of the spindle envelope, but this could be clarified.

Answer: Correct, the vertical lines indicate the onset of spindle (envelope). We now clarified this in the figure legend.

3. Also related to figure 3a, why do the spectrograms show a prominent narrow band of slow wave activity around 3 Hz for TRN and VB, whereas it is much lower (0-1 Hz) in most of the other areas examined, including other thalamic areas AD and CMT? Is there something different about the slow waves in these areas?

Answer: This is an interesting point that also surprised us. Our current explanation is that these different local oscillations likely reflect different thalamocortical and cortico-thalamic circuits. The dominant frequency may directly relate to the connectivity and firing of the local underlying circuitries. Although we have not investigated this aspect in the present study, previous work from our lab (Gent, Bandarabadi et al. 2018) and others (Huber, Ghilardi et al. 2004, Vyazovskiy, Olcese et al. 2011) have shown that slow waves activities are locally regulated and strongly dependent on the site of recording and previous activity/experience (e.g., repeated motor task or arm immobilization (Huber, Ghilardi et al. 2006). For instance, our previous work showed that after prolonged period of wakefulness (typically at the end of a sleep deprivation procedure), when animals are mostly awake, slow wave activities are high in almost all prefrontal cortex and corresponding medio-dorsal thalamic nuclei, while completely absent in sensory thalamus such as VB nucleus (Gent, Bandarabadi et al. 2018). Consistent with this view, these local oscillations may reflect two concurrent oscillations in the delta band, namely delta 1 or slow wave (<1.5 Hz) and delta 2 (1.5-4 Hz), as suggested by (Hubbard, Gent et al. 2019).

4. The figure 3 legend lists a, b, c, and d panels, but only a-c are in the actual figure.

Answer: We corrected the figure legend accordingly.

5. In the figure legend for Figure 4b, can the authors clarify that the brown-shaded areas in the top tracing are expanded in middle tracings below, with perhaps better labeling of the pre-stim vs intra-stim tracings. Also, I assume the red arrows are highlighting spindles, but this could be stated explicitly.

Answer: Correct, we clarified the figure legend accordingly.

6. In the recovery sleep experiments, the authors note that spindle density is increased. In the context of recovery sleep, do some of the earlier findings still apply? i.e. is spindle density equally elevated before

REM during recovery sleep? Is the increased spindle density during recovery sleep predictive of longer REM bouts?

Answer: Those are interesting points. We performed new analysis on our datasets, and indeed, we found that the increase of spindle density before REM during sleep rebound is (even) significantly higher than during baseline (now included in Fig. 4c).

In addition, we found no correlation between REM length and spindle rate when calculated from the entire NREM episode duration (N2R) or the 25 s before NREM-to-REM sleep transitions (Supp. Fig. 5), further supporting an implication of spindle in NREM-REM transitions.

7. In the Methods under Stereotaxic Injection of AAV, there is no mention of the step function opsin and its control (only ChETA). Also include virus titer and total volume injected.

Answer: We added these experimental details to the method section.

8. In the legend for Figure 2, can the authors reiterate the number of animals that contributed to the unit pooling?

Answer: We corrected the figure legend accordingly.

9. The authors used only male mice in these experiments. Given that females have increased spindle density compared to males, a discussion of the generalizability and potential effect of sex may be warranted.

Answer: This is a very good point that is now discussed in the discussion section.

10. The discussion of and potential explanation for reversed spike-field coupling within the BARR cortex is appreciated. A study in naturally sleeping rats found neocortical units fired at the trough of a spindle (trough-locked) which corresponded to a layer 4 sink and layer 5 source (Sirota et al, PNAS 2003), and similarities/differences to this finding may warrant discussion.

Answer: We appreciate the reviewer's inputs here and agreed that difference in electrodes placement may be the reason for this reversed spike-field coupling, although all this was observed in all our animals. We discussed this accordingly in the revised discussion section.

Reviewer #2

The article by Bandarabadi, Gutierrez Herrera et al. describes properties of spindle oscillatory patterns in different thalamic nuclei and the neocortex in mice. With an innovative optogenetic stimulation protocol in TRN, they increase spindling during NREM sleep and find that this prolongs later REM sleep periods. I enjoyed reading this paper. It argues well why multisite recordings in freely behaving mice, and naturalistic spindle stimulation protocols add value to the extant spindle related research (e.g. p. 19 l. 423ff). It was, however, hard for me to follow why this particular set of results was presented in one manuscript and how the different aspects contribute to a main takeaway. The paper reports two sets of results, (1) spindle detection and properties in different thalamic nuclei and the neocortex. (2) The relation of spindling to sleep architecture (and effects of spindle stimulation on sleep architecture). It did not become entirely clear to me, why (1) is necessary to report, and how it supports the findings relating to (2). In my opinion, how spindles relate and maybe control sleep architecture is the more novel and exciting finding reported here. I therefore believe that this manuscript could be made a stronger paper if the first part was e.g. moved to the supplement (if the authors feel the need to report evidence that validates their spindle detection method), or considerably shortened, and the authors focused instead on reporting results relating to objective 2). This would additionally give them more space for better theoretical embedding of that topic.

We would like to thank the reviewer for the supportive comments. As stated above, we have decreased the section on spindles detection and characterization and have improved our functional investigation of the role of spindles on NREM-REM sleep transitions by adding new experiments and datasets.

Comments:

1. A lot of space is given to description of the spindle detection method. It is not clear, however, whether the differences in this algorithm (wavelet based, and specifically the B-spline function) crucially contributed to the new findings, or whether the same pattern of results would have been observed with other methods? I personally think that robustness towards different analysis methods speaks to the robustness of an observed effect. It might not be necessary to stress the specifics of the used method in a lot of detail, if it is not at the center of the findings. I would find it helpful if the authors commented in the description of their method whether it merely detects spindles more precisely, or whether the detection method was actually crucial to their findings.

Answer: We understand the reviewer's point, and it is likely that a similar set of qualitative results would have been obtained, however we stress the importance of our detection method, and its versatility, in the quantitative analysis of spindles in rodents (specifically the B-Spline function) where spindles are often detected using a simple threshold analysis. However, detection of spindles in mice, unlike in human EEG recordings, is a tricky task and would need optimization, which we first investigated in this study. Using a threshold analysis, we would not have obtained a precise quantification of the local distribution of spindles in the thalamus (intra-cranial electrodes/tetrodes) or spindle rates across baseline and sleep deprivation (from EEG electrodes).

In support of this, we now compared the sensitivity of our method with the wavelet-based method with the complex Morlet function and showed that our method detects spindles more precisely, including a low false detection rate (FDR; Fig. 1d).

Furthermore, to put less stress on the methodology part, we downsized the description of the spindle detection method in the main text, and moved some parts to the Method section.

2. What were the controls for the optogenetic stimulation results? How exactly were the baseline values obtained? Can the authors rule out non-specific effects of light, heat, or protein expression?

Answer: This is a valid point. Previously published experiments from our group showed that light, heat, or protein expression did not show non-specific effects on the observed circuit dynamics and behavioral switches reported (Jego, Glasgow et al. 2013, Boyce, Glasgow et al. 2016, Herrera, Cadavieco et al. 2016, Gent, Bandarabadi et al. 2018).

Nevertheless, we are now including new experiment implicating the TRN-AD circuit in NREM-to-REM sleep transitions. In this new set of experiment, we included a YFP control that address the concerns of the reviewer about non-specificity of the stimulation. As expected, optogenetic control experiment did not result in a significant change of sleep-wake architecture or spindle occurrence (see Fig. 3d,e).

3. Does the optogenetic stimulation protocol allow testing whether stimulation and thus increased spindling can promote/facilitate REM sleep onset as suggested by the authors given the results reported on spindle grouping relative to vigilance state transitions? This would help answer whether spindles have a causal role here or whether the timing of REM sleep is controlled by the infraslow oscillations that in turn control frequency of spindle occurrence.

Answer: The short answer is “Yes it does” and it is the core finding of this study. Indeed, our optogenetic activation of TRN^{GABA} neurons induced a significant increase of NREM-to-REM sleep transitions (from ~30 % to ~80%, see Fig. 3e), therefore establishing a causal link between TRN cell activity, spindles occurrence and the probability of REM sleep transition.

Whether spindles (direct) or infra-slow oscillations (indirect) are responsible in the observed REM sleep transitions remains an open question. Our work suggest that spindles are an essential element in these transitions, but we didn't test directly the effect of infra-slow oscillations since their mechanisms remain elusive. Our view on this question is that infra-slow oscillations may reflect the firing activity of TRN cells, and therefore spindle rate, linking the two phenomena to NREM sleep stabilization and REM sleep transitions form NREM sleep. Yet, this remains to be experimentally tested with long-duration manipulation of TRN cell firing.

Finally, we now provide novel experiment using optogenetic activation of the TRN-AD circuit that causally implicate spindle in NREM-to-REM transitions (see fig 3d,e). These results further confirmed our statement about a causal link between spindles and REM sleep. We have improved the discussion of this point in the revised manuscript.

4. Spindle results in BARR: It could be made even more prominent that the inverse findings in BARR probably relate to the fact that an unusual layer was recorded, such that readers do not put too much focus/weight on the directionality of these findings (e.g. p. 8 l.185).

Answer: We agreed with the reviewer and corrected the revised manuscript accordingly. The corresponding text in the discussion has been shortened.

5. p.17 l.364 confirmed that TRN is a driver of spindles is a little misleading in context. The authors only show this for the classical TRN-VB-BARR pathway, not for the CMT and AD pathways the report as novel findings in part 1).

Answer: We apologized for this overstatement and agreed with the reviewer's comments since spindle can also be generated by activation of cortico-thalamic inputs (Bonjean, Baker et al. 2011). We have corrected this sentence in the revised manuscript. Importantly, we now provide additional experiment showing that optogenetic activation of the TRN-AD circuit is sufficient to increase spindles (see Fig. 3) and to promote NREM-to-REM sleep transitions. This finding strengthens our previous results and emphasize the fact that several spindle generating circuits may exist in the thalamus. This is now discussed in the revised discussion.

6. What is a possible significance of increased spindle rate during recovery sleep? How does this finding relate to the other results reported in this paper? It was hard for me to follow how this adds to the authors' message, it would be nice if more context were given.

Answer: We apologize for the lack of clarity on this point. These findings relate directly to the implication of spindles in sleep stability, both NREM sleep and the following REM sleep. Furthermore, an increase probability of transition to REM sleep ultimately facilitate "complete" sleep cycle (NREM-REM) for optimal recovery.

7. What is a possible significance of the additional spindle pathways (CMT, AD) compared to the classical TRN VB? Discussion of this/directions for further research would strengthen the case of including the results.

Answer: This is an interesting point that we have not developed much in the previous version of the MS. Anatomically, sensory thalamic nuclei (VB) have no direct or indirect connections with the CMT, which is a non-specific midline thalamus. Yet, our study clearly shows the existence of spindles in these nuclei (CMT, AD). Therefore, strong spike-field coupling within these nuclei suggests a separate circuit for spindle generation that may generate spindles independent of the classical TRN-VB-Barrel cortex circuit. Spindles occurring in these separate networks may have different functional significance that directly relate to the region-specific organization of sleep structure sleep functions, including the coordination of somatosensory (VB) information and attention or executive functions (CMT, AD).

8. Have the authors considered mu motor rhythm during wakefulness and alpha bursts during REM sleep as possible contributors to “falsely” detected spindle events in wake and REM where sleep spindles in the traditional sense are not expected?

Answer: This is a very relevant point; however, there is no experimental proof of the existence of “mu motor rhythm” in rodents, as described in human. In contrast, although there is no “alpha burst” during REM sleep in rodent (again, this is an EEG features recorded mainly during sleep in human), there is a theta burst/rhythmic theta/high theta during REM sleep (Montgomery, Sirota et al. 2008), which our algorithm detect during REM sleep (see supp. Fig. 2). Those were identified as spindle-like events during wakefulness and REM sleep and excluded from our analysis.

Additional notes:

1. Note of caution: I would suggest using frequency of spindle occurrence rather than spindle frequency. Particularly in the human literature (which is also cited in the introduction) spindle frequency usually relates to the oscillatory frequency of a spindle (as in fast and slow spindles). When talking about both in the same manuscript, explicit specification may avoid confusion.

Answer: We now used the “spindle rate” to refer to frequency of spindle occurrence, and “central frequency” to refer to the central oscillatory frequency of spindles.

2. Sometimes, small inconsistencies in language made it difficult for me to follow the authors line of argumentation (e.g. p. 8 180-182, “but significantly less than of TRN”, p. 10 l.222-225, p. 17 l.365-367 “... which increases in frequency”).

Answer: We corrected those and other sentences in the revised MS.

Reviewer #3

The authors present an impressive and novel paper showing that 1) areas of the thalamus that have not previously been linked with spindles show spindle coupling, and 2) spindles increase briefly before REM sleep, and stimulating spindles increases the length of ensuing REM. The paper is very well-written, statistically thorough, and the literature is well-covered. I do believe, however, that the paper could be improved by considering the points below.

Answer: We thank the reviewer for the supportive comments.

Major considerations:

1. My largest issue is not one due to faulty methods or incorrect interpretations. It's simply that this paper feels like it has two separate ideas that do not connect well. The first section talks about evidence that thalamic nuclei that have not been previously linked with spindles show spindle coupling; the second section uses optogenetics to stimulate spindles and shows an interesting connection with REM length as a result. But the authors do not record from (or do not report on) the same non-specific areas in the second section, making me wonder why these are not separate papers. To be clear, I believe the second section is interesting enough on its own to warrant publication here (provided the other reviewers do not raise substantial concerns), whereas the first is more fitting for a more specialized journal, considering there is no novel contribution of what function these areas support. If the authors prefer to keep these as the same paper, I'm wondering if they could skirt through the first section more quickly and expand upon the second section.

Answer: We agreed with the reviewer statement and revised our manuscript content and structure accordingly. Please see our "revision summary" above. We hope these modifications will satisfy the reviewer.

2. The optogenetic stimulations occurred every 10 seconds. One way to expand the second section is to ask more questions related to these stimulations, such as - were the stimulations any less effective at increasing spindles when they occurred shortly after preceding spindles (when the system has been proposed to be in refraction)? That is, do they differ as a function of the time since the preceding spindle? This could be important for understanding the characteristics of this refractory period and, in turn, spindle generation.

Answer: There might be a slight misunderstanding here. To clarify, the step functions opsin used here is quite different from the classical ChR2. While the latter one require continuous light stimulation to be active (open state, hence, inducing action potential), SSFO remains open after a single light pulse and slowly close over dozens of seconds to minutes during which we delivered single light pulse every 10s to keep them open (we close them with a flash of yellow/green light). Thus, SSFO enables an overall increase of TRN cell excitability and firing, in particular in burst mode during NREM sleep, together with an increase in spindle occurrence. SSFO do not allow to elicit single spindles with millisecond timescale. Therefore, we do not elicit spindles with single light pulses, as assumed by the reviewer, but rather increase their general occurrence when the SSFO channel is open.

3. It appears stronger coupling between slow waves and spindles occurs in AD, CMT, and CING because of stronger slow wave power in these regions, as seen in figure 3A. The authors also briefly touch on this idea in the discussion without testing it explicitly. Could the authors verify (via quantification) whether this is indeed what drives the coupling difference?

Answer: This is an interesting point. We performed new analysis and calculated delta power (0.5-4 Hz) during spindles, and then estimated correlation between SW-spindle coupling and delta power during spindles (Fig. 5d). We found a positive correlation for the cortical sites, but not for the thalamic recordings, suggesting that other mechanism may be involved.

4. Regarding the increase in spindles after sleep loss, it is well-known that slow waves increase after sleep loss. Is the spindle increase explained by this change in slow waves, or is it meaningfully independent?

Answer: In a new data analysis, we estimated delta power during spindles within first 30 minutes of recovery sleep (Fig. 5e). We found that delta power significantly increases for cortical LFP recordings, but not for thalamic ones, during spindles of recovery sleep. Together with results of SW-spindle coupling, these suggest that SW power can modulate both incidence and SW-spindle coupling in the cortical sites, but not in the thalamic nuclei. We included these new analysis and results in the revised manuscript and discussed them accordingly.

5. In figure 5C, it is not clear what the baseline control values are for the first 40 or 60 minutes, and how this recovery night comparison to the baseline as a result. Please either plot this as a different score or plot the baseline night as well.

Answer: They are now normalized to their baseline values in the updated figure.

Minor considerations:

1. One impressive and understated aspect of this paper is the development of a new, superior spindle algorithm. If they have not already, the authors should consider sharing the code to this algorithm as part of the paper.

Answer: Thanks for highlighting this. We will share scripts of this study as a part of this paper, as also has been requested by the publisher.

2. The author state, "Our findings, thus, provide causal evidence for the extension of the classical somatosensory TRN-VB-BARR pathway for propagation of spindles within thalamocortical circuits." Could the authors please explain how this claim is justified?

Answer: Thanks for raising this point. A strong spike-field coupling within CMT and AD demonstrates regional cellular substrates of local spindles, as the LFP measures the synaptic inputs of a large number of neurons (Buzsaki, Anastassiou et al. 2012), thus indicates that the activity of CMT and AD neurons are

entrained during spindles. This only happens when these neurons receive synaptic inputs during spindles, and therefore are part of spindle pathway. This claim is now substantiated by a set of novel experiments implicating the TRN^{GABA}-AD circuits in spindle generation (Fig. 3d,e).

3. The author state, "The natural extension to this is that mice do not have centro-parietal spindles, i.e. predominantly TRN originating spindles." Could the authors please explain what they mean by this, using references to back up their claim?

Answer: We reckon the strong tone of this claim and removed it in the revised manuscript, based on our novel (additional) finding of the implication of TRN-AD circuit in non VB-BARR spindle generation.

4. Please fix this sentence: "We found that spindles waveforms were highly symmetrical in all the recorded sites."

Answer: we corrected it.

5. Please fix this sentence: "The complex frequency B-Spline wavelet function has significantly a higher normalized power compared to other functions (frequency B-spline versus other functions: $P < 0.0001$; $F = 14.22$; d.f. = 14; one-way ANOVA; $n = 6$ subjects)."

Answer: we corrected it.

References

- Bonjean, M., T. Baker, M. Lemieux, I. Timofeev, T. Sejnowski and M. Bazhenov (2011). "Corticothalamic feedback controls sleep spindle duration in vivo." *J Neurosci* **31**(25): 9124-9134.
- Boyce, R., S. D. Glasgow, S. Williams and A. Adamantidis (2016). "Causal evidence for the role of REM sleep theta rhythm in contextual memory consolidation." *Science* **352**(6287): 812-816.
- Buzsaki, G., C. A. Anastassiou and C. Koch (2012). "The origin of extracellular fields and currents--EEG, ECoG, LFP and spikes." *Nat Rev Neurosci* **13**(6): 407-420.
- Devuyst, S., T. Dutoit, P. Stenuit and M. Kerkhofs (2011). "Automatic sleep spindles detection--overview and development of a standard proposal assessment method." *Conf Proc IEEE Eng Med Biol Soc* **2011**: 1713-1716.
- Gent, T. C., M. Bandarabadi, C. G. Herrera and A. R. Adamantidis (2018). "Thalamic dual control of sleep and wakefulness." *Nat Neurosci* **21**(7): 974-984.
- Herrera, C. G., M. C. Cadavieco, S. Jego, A. Ponomarenko, T. Korotkova and A. Adamantidis (2016). "Hypothalamic feedforward inhibition of thalamocortical network controls arousal and consciousness." *Nat Neurosci* **19**(2): 290-298.
- Hubbard, J., T. C. Gent, M. M. B. Hoekstra, Y. Emmenegger, V. Mongrain, H.-P. Landolt, A. R. Adamantidis and P. Franken (2019). "Reassessing the validity of slow-wave dynamics as a proxy for NREM sleep homeostasis." *bioRxiv*: 748871.

Huber, R., M. F. Ghilardi, M. Massimini, F. Ferrarelli, B. A. Riedner, M. J. Peterson and G. Tononi (2006). "Arm immobilization causes cortical plastic changes and locally decreases sleep slow wave activity." Nat Neurosci **9**(9): 1169-1176.

Huber, R., M. F. Ghilardi, M. Massimini and G. Tononi (2004). "Local sleep and learning." Nature **430**(6995): 78-81.

Jego, S., S. D. Glasgow, C. G. Herrera, M. Ekstrand, S. J. Reed, R. Boyce, J. Friedman, D. Burdakov and A. R. Adamantidis (2013). "Optogenetic identification of a rapid eye movement sleep modulatory circuit in the hypothalamus." Nat Neurosci **16**(11): 1637-1643.

Montgomery, S. M., A. Sirota and G. Buzsaki (2008). "Theta and gamma coordination of hippocampal networks during waking and rapid eye movement sleep." J Neurosci **28**(26): 6731-6741.

Vyazovskiy, V. V., U. Olcese, E. C. Hanlon, Y. Nir, C. Cirelli and G. Tononi (2011). "Local sleep in awake rats." Nature **472**(7344): 443-447.

REVIEWERS' COMMENTS:

Reviewer #1 (Remarks to the Author):

I appreciated what the authors did to streamline and connect the manuscript, and while the flow is improved, the spindle coupling findings and REM influencing findings still comes across as a set of loosely tied observations.

It seems to bear noting that one of the primary observations of the earlier version of the manuscript – that increasing excitability of the TRN using optogenetics leads to prolonged subsequent REM sleep duration (prior figure 4E) is no longer true. Instead, the authors find that the probability of transition from NREM to REM (rather than Wake) is now significantly increased, without impacting the latency of transition to REM, or (in the case of stimulation of the TRN) impacting REM bout length. First, I think it is important to use clear language about this finding. I recommend changing the section heading for these results to: "Driving TRN-GABA cells or their projections to AD increases spindle rate and the probability of transition to REM sleep" since otherwise saying "...increases transitions to REM sleep" makes it sound like the raw number of transitions to REM sleep in the experimental condition over some long duration of sleep recording was increased compared to the control condition, which, if I understand the experimental paradigm correctly, was not evaluated. This consistent language should be used throughout the Results and Discussion.

Second, there seems to be a dichotomy between the effect of driving TRN-GABA cells and their projections to AD: both appear to increase the probability of transition to REM sleep, but whereas the former has no effect on REM sleep latency, the latter actually significantly *decreases* subsequent REM sleep latency. This seems to be an important observation and has implications about output specificity of the TRN and how REM sleep is regulated that would seem to warrant greater discussion.

Minor Points:

1. For Figures 3D and 3E, I prefer the data presentation as it is in Figure 3C (i.e. individual data points overlaid on the bars).
2. In Figure 3D, the EEG leads are referred to as "Front." and "Occip." but elsewhere as EEG1 and EEG2. I prefer the "Front." and "Occip." nomenclature, but in any case the naming should be consistent.
3. In lines 281-284 is the sentence, "In addition, we found no correlation between REM bout duration and spindle rate when calculated for the entire NREM episode duration (N2R) or the 25 s before NREM-to-REM sleep transitions (Supplementary Fig. 5), further supporting an implication of spindles in NREM-REM transitions." However this observation does not seem to support a role for spindles in NREM-REM transitions, so the final clause should probably be left off.
4. Throughout several places in the manuscript, thalamic nuclei such as the AD and CMT are referred to as "non-specific." I think the authors might mean these are "non-sensory" nuclei, since one could argue that their individual functional roles and connectivity patterns would qualify them as specific.
5. In response to prior Minor Point 7, the authors indicate that viral titer and total volume injected of the AAV encoding the step-function opsin were added to the Methods, but I don't see that this has been done.

The authors otherwise satisfactorily addressed the other points in my initial review.

Reviewer #2 (Remarks to the Author):

I thank the authors for the thorough responses to my comments and commend them for running

an additional experiment that provides causal evidence that spindles are involved in the regulation of subsequent REM sleep. Together with the changes to the manuscript that logically tie the previously separate parts, the readability of the paper as a whole has improved!

All of my previous comments have been answered satisfactorily.

Minor comments on the revised version:

II. 286 f please correct sentence: "An increased power in delta band activity one of the major features of the sleep homeostatic process that compensate for sleep loss⁵²."

II. 389 ff please correct grammar: "We further showed that optogenetic driving of TRNGABA neurons, or TRNGABA-AD projections, are sufficient to induce spindles and promote NREM to REM sleep transitions."

II. 457 f please clarify "Anatomically, sensory thalamic nuclei (VB) have no direct connections with the CMT, which is a non-specific midline thalamus.

Reviewer #3 (Remarks to the Author):

I congratulate the authors on a successful revision, including informative new data, clarifying analyses, and improvements in theoretical considerations. I feel pretty happy with the aggregate results and related discussion.

However, it seems the results section could have a better flow to convey its ideas. One idea would be to re-organize it into something like the following, using the current figure numbers: Fig 1, Fig 2, Fig 5, Fig 4, Fig 3. Fig 1, 2, & 5 describe the method, show spindles occur in these new regions, and show spindles are modified by slow waves in each of these regions, which grounds the method in the prior literature and extends it to these new regions and how slow waves are involved. Fig 4 shows that spindles signal a transition to REM, and then Fig 3 builds on this by showing causal evidence that this is the case. The relationship between Fig 4 & 3 to the earlier parts of the paper would be natural, because of the new TRN-AD stimulation experiment – not only do the optogenetic experiments solidify the importance of spindles in precipitating REM sleep, but they also show these stimulations bring about REM sleep when stimulating regions that were first brought to relevance (with respect to spindles) in the prior section of this paper. The authors could introduce recovery sleep in the slow wave section (since that is the most discussed prior physiological change during recovery) and continue to mention it in talking about spindles reflecting a transition to REM. To me, that seems the most natural way to organize the paper using the same results. As it stands, I feel the most exciting part of the results is in the middle (Fig 3), and the paper kind of strays theoretically / does not build on itself in a straightforward way.

Another consideration would be to remove or place the slow wave section and recovery sleep experiments (including part of the current Fig 4) in the supplementary material. Neither are necessary for the main conclusions of the paper, and this would focus the paper on their two main findings: 1) extending spindle findings to new regions and 2) showing causal evidence that stimulating well-established (TRN) and novel (TRN-AD) spindle regions brings about REM.

I do not think the authors MUST do either of these things to publish the paper, but I do think they would strengthen it.

Minor point: Please fix this sentence: "The complex frequency B-Spline wavelet function has significantly a higher normalized power as to other functions".

Reviewer #1 (Remarks to the Author):

I appreciated what the authors did to streamline and connect the manuscript, and while the flow is improved, the spindle coupling findings and REM influencing findings still comes across as a set of loosely tied observations.

It seems to bear noting that one of the primary observations of the earlier version of the manuscript – that increasing excitability of the TRN using optogenetics leads to prolonged subsequent REM sleep duration (prior figure 4E) is no longer true. Instead, the authors find that the probability of transition from NREM to REM (rather than Wake) is now significantly increased, without impacting the latency of transition to REM, or (in the case of stimulation of the TRN) impacting REM bout length. First, I think it is important to use clear language about this finding. I recommend changing the section heading for these results to: “Driving TRN-GABA cells or their projections to AD increases spindle rate and the probability of transition to REM sleep” since otherwise saying “increases transitions to REM sleep” makes it sound like the raw number of transitions to REM sleep in the experimental condition over some long duration of sleep recording was increased compared to the control condition, which, if I understand the experimental paradigm correctly, was not evaluated. This consistent language should be used throughout the Results and Discussion.

Answer: In the previous version of the manuscript, we had analyzed the whole recordings of the optogenetic experiments, including the periods of recording without stimulation. In the revised version, and based on other reviewer’s comments, we refined the quantifications to the periods of recording during optogenetic stimulation to have a proper estimation of their effects on sleep-wake transitions. We now clarified this in the final version of the manuscript. We also updated the section heading per reviewer’s suggestion.

Second, there seems to be a dichotomy between the effect of driving TRN-GABA cells and their projections to AD: both appear to increase the probability of transition to REM sleep, but whereas the former has no effect on REM sleep latency, the latter actually significantly *decreases* subsequent REM sleep latency. This seems to be an important observation and has implications about output specificity of the TRN and how REM sleep is regulated that would seem to warrant greater discussion.

Answer: Our data indeed show that activation of either TRN^{VGAT} cells and their projections to AD all increase the probability of transition to REM sleep. However, there is no significant difference on the latency to REM sleep. Only the latency of NREM sleep to wake (N2W) is decreased, possibly due to a secondary effect due to the increase N2R. Although our data

do not support such dichotomic effects, we emphasized the possible function of these circuits (TRN-AD) in the discussion section.

Minor Points:

1. For Figures 3D and 3E, I prefer the data presentation as it is in Figure 3C (i.e. individual data points overlaid on the bars).

Answer: We updated the figure per reviewer's suggestion, as it also was suggested in the editorial comments.

2. In Figure 3D, the EEG leads are referred to as "Front." and "Occip." but elsewhere as EEG1 and EEG2. I prefer the "Front." and "Occip." nomenclature, but in any case the naming should be consistent.

Answer: We now corrected this and used consistent terms across manuscript/figures.

3. In lines 281-284 is the sentence, "In addition, we found no correlation between REM bout duration and spindle rate when calculated for the entire NREM episode duration (N2R) or the 25 s before NREM-to-REM sleep transitions (Supplementary Fig. 5), further supporting an implication of spindles in NREM-REM transitions." However this observation does not seem to support a role for spindles in NREM-REM transitions, so the final clause should probably be left off.

Answer: We corrected the sentence as follow "In addition, we found no correlation between REM bout duration and spindle rate when calculated for the entire NREM episode duration (N2R) or the 25 s before NREM-to-REM sleep transitions (Supplementary Fig. 5), further supporting an implication of spindles in NREM-REM transitions, rather than REM sleep duration."

4. Throughout several places in the manuscript, thalamic nuclei such as the AD and CMT are referred to as "non-specific." I think the authors might mean these are "non-sensory" nuclei, since one could argue that their individual functional roles and connectivity patterns would qualify them as specific.

Answer: Agree. We updated the term across the manuscript.

5. In response to prior Minor Point 7, the authors indicate that viral titer and total volume injected of the AAV encoding the step-function opsin were added to the Methods, but I don't see that this has been done.

Answer: We now included the values.

The authors otherwise satisfactorily addressed the other points in my initial review.

Reviewer #2 (Remarks to the Author):

I thank the authors for the thorough responses to my comments and commend them for running an additional experiment that provides causal evidence that spindles are involved in the regulation of subsequent REM sleep. Together with the changes to the manuscript that logically tie the previously separate parts, the readability of the paper as a whole has improved!

Answer: We thank the reviewer for the support.

All of my previous comments have been answered satisfactorily.

Minor comments on the revised version:

II. 286 f please correct sentence: "An increased power in delta band activity one of the major features of the sleep homeostatic process that compensate for sleep loss⁵²."

Answer: We have corrected it.

II. 389 ff please correct grammar: "We further showed that optogenetic driving of TRNGABA neurons, or TRNGABA-AD projections, are sufficient to induce spindles and promote NREM to REM sleep transitions."

Answer: We have corrected it.

II. 457 f please clarify "Anatomically, sensory thalamic nuclei (VB) have no direct connections with the CMT, which is a non-specific midline thalamus.

Answer: We have corrected it.

Reviewer #3 (Remarks to the Author):

I congratulate the authors on a successful revision, including informative new data, clarifying analyses, and improvements in theoretical considerations. I feel pretty happy with the aggregate results and related discussion.

However, it seems the results section could have a better flow to convey its ideas. One idea would be to re-organize it into something like the following, using the current figure numbers: Fig 1, Fig 2, Fig 5, Fig 4, Fig 3. Fig 1, 2, & 5 describe the method, show spindles occur in these new regions, and show spindles are modified by slow waves in each of these regions, which grounds the method in the prior literature and extends it to these new regions and how slow waves are involved. Fig 4 shows that spindles signal a transition to REM, and then Fig 3 builds on this by showing causal evidence that this is the case. The relationship between Fig 4 & 3 to the earlier parts of the paper would be natural, because of the new TRN-AD stimulation experiment – not only do the optogenetic experiments solidify the importance of spindles in precipitating REM sleep, but they also show these stimulations bring about REM sleep when stimulating regions that were first brought to relevance (with respect to spindles) in the prior section of this paper. The authors could introduce recovery sleep in the slow wave section (since that is the most discussed prior physiological change during recovery) and continue to mention it in talking about spindles reflecting a transition to REM. To me, that seems the most natural way to organize the paper using the same results. As it stands, I feel the most exciting part of the results is in the middle (Fig 3), and the paper kind of strays theoretically / does not build on itself in a straightforward way.

Answer: Thanks for the suggestion. We updated the order of figures and results section in the final version of the manuscript per your suggestion.

Another consideration would be to remove or place the slow wave section and recovery sleep experiments (including part of the current Fig 4) in the supplementary material. Neither are necessary for the main conclusions of the paper, and this would focus the paper on their two main findings: 1) extending spindle findings to new regions and 2) showing causal evidence that stimulating well-established (TRN) and novel (TRN-AD) spindle regions brings about REM.

Answer: We believe that dynamics of spindles before transition to REM sleep during recovery sleep is an important finding, thus, we would like to keep it in the main manuscript.

I do not think the authors MUST do either of these things to publish the paper, but I do think they would strengthen it.

Minor point: Please fix this sentence: "The complex frequency B-Spline wavelet function has significantly a higher normalized power as to other functions".

Answer: We have corrected it.